


# The 1-way on-line coupled model system MECO(n) – Part 4: Chemical evaluation (based on MESSy v2.52)

Mariano Mertens[1], Astrid Kerkweg[2], Patrick Jöckel[1], Holger Tost[2], and Christiane Hofmann[2]

[1]Deutsches Zentrum für Luft- und Raumfahrt, Institut für Physik der Atmosphäre, Oberpfaffenhofen, Germany
[2]Institut für Physik der Atmosphäre, Johannes Gutenberg-Universität Mainz, 55099 Mainz, Germany

*Correspondence to:* Mariano Mertens (mariano.mertens@dlr.de)

**Abstract.** For the first time a simulation incorporating tropospheric and stratospheric chemistry using the newly developed MECO(n) model system is performed. MECO(n) is short for MESSyfied ECHAM and COSMO model nested n-times. It features an on-line coupling of the COSMO-CLM model, equipped with the Modular Earth Submodel System (MESSy) interface (called COSMO/MESSy), with the global atmospheric chemistry model ECHAM5/MESSy for Atmospheric Chemistry (EMAC). This on-line coupling allows a consistent model chain with respect to chemical and meteorological boundary conditions from the global scale down to the regional kilometre scale.

A MECO(2) simulation incorporating one regional instance over Europe with 50 km resolution and a one instance over Germany with 12 km resolution is conducted for the evaluation of MECO(n) with respect to tropospheric gas-phase chemistry. The main goal of this evaluation is to ensure, that the chemistry related MESSy submodels and the on-line coupling with respect to the chemistry are correctly implemented. This evaluation is a prerequisite for the further usage of MECO(n) in atmospheric chemistry related studies. Results of EMAC and the two COSMO/MESSy instances are compared with satellite-, ground-based- and aircraft in situ observations, focusing on ozone, carbon monoxide and nitrogen dioxide. Further the methane lifetimes in EMAC and the two COSMO/MESSy instances are analysed in view of the tropospheric oxidation capacity. From this evaluation we conclude that the chemistry related submodels and the on-line coupling with respect to the chemistry are correctly implemented. In comparison with observations both, EMAC and COSMO/MESSy, show strengths and weaknesses. Especially in comparison to aircraft in situ observations COSMO/MESSy shows very promising results. However, the amplitude of the diurnal cycle of ground-level ozone measurements is underestimated. Most of the differences between COSMO/MESSy and EMAC can be attributed to differences in the dynamics of both models, which is subject to further model developments.

## 1 Introduction

The emissions of reactive compounds are a key component for the simulation of atmospheric chemistry processes. Many of these emissions are localised as for example along ship tracks or highways. Especially, as some of the relevant processes (for example tropospheric ozone chemistry) are non-linear, it is desirable to resolve smaller scales, since with finer resolution the capabilities of chemistry-climate models in simulating species like ozone or nitrogen dioxide can be enhanced. In addition, a finer resolution can improve the ability to represent the physical processes and the dynamics. The resolution of global



chemistry-climate models, however, can only be increased to a certain degree, as current computational resources pose an upper limit. Therefore the new model system **ME**SSyfied **E**CHAM and **CO**SMO model nested **n**-times (MECO(n)) has been developed. This system includes the regional scale chemistry-climate model COSMO-CLM/MESSy (from now on denoted as COSMO/MESSy), i.e. an implementation of the **M**odular **E**arth **S**ubmodel **Sy**stem (MESSy, Jöckel et al., 2005) framework into

the regional weather prediction and climate model of the **CO**nsortium for **S**mall-scale **MO**deling (COSMO, Doms and Schät­tler, 2002, Steppeler et al., 2003; COSMO-CLM, Rockel et al., 2008) as described in detail by Kerkweg and Jöckel (2012a). Additionally, the preprocessing tool "INT2LM", which is provided by the German Weather Service (DWD) for the calculation of the initial and boundary data of the regional COSMO model was implemented into MESSy as submodel INT2COSMO by Kerkweg and Jöckel (2012b).

The implementation of the MESSy infrastructure in COSMO (including INT2COSMO) allows for an on-line coupling between the different MECO(n) instances. This means that individual COSMO/MESSy instances can be driven on-line from the global chemistry-climate model **E**CHAM5**/MESS**y for **A**tmospheric **C**hemistry (EMAC, Jöckel et al., 2006, 2010) or from coarser resolved COSMO/MESSy instances. Especially for complex chemistry climate applications with several hundreds of different tracers, this on-line nesting is a key advantage of MECO(n) compared to the traditional off-line nesting. There

is no need to store information for the boundary conditions on disk, instead they are interchanged using a point-to-point communication based on the message passing interface (MPI). This direct exchange of boundary conditions allows for a much higher update frequency of the boundary conditions, as new data are provided at each time step of the driving model.

A second benefit is the consistence of the boundary and initial data between the driving model and the regional refinement, as the same chemical set-up can be used in all instances. Comparable model systems without on-line nesting of the regional

refinement often use constant chemical boundary conditions (WRF/Chem, Grell et al., 2005) or use results from global models like MOZART (e.g. COSMO-ART, Knote et al., 2011 or WRF/Chem, Žabkar et al., 2015). In these cases not only the update frequency is limited, but also the chemical speciation between driving and regional model might be different. Due to different chemical speciation between the driving model and the regional model (or if realistic boundary conditions are completely lacking) additional biases can be introduced. In addition the meteorological and chemical fields applied as boundary conditions

might be inconsistent as in many application they stem from different models.

In the traditional off-line nesting approach COSMO(-CLM) is usually driven by reanalysis data. In the case of MECO(n) however, COSMO/MESSy is driven by the meteorological fields provided by EMAC. By this, biases are potentially introduced, which might have a negative impact on the quality of the meteorological conditions.

To test this, Hofmann et al. (2012) compared results from the classical off-line nested version of the COSMO model (using

ECMWF analysis data) to results from the on-line nested set-up with EMAC nudged towards the same analysis data. It was shown that both approaches for all three cases (a cold front, a convective frontal event and a winter storm) lead to results with a comparable accuracy between the on-line and off-line nesting.

Nevertheless, before MECO(n) can be used with a complex chemistry set-up for atmospheric chemistry studies, it is crucial to evaluate the model performance with respect to gas-phase chemistry. For this reason, this paper is dedicated to the chemical

evaluation of MECO(n) with focus on tropospheric gas-phase chemistry. Our goal is to test the implementation of the chemical



processes and the on-line coupling of the chemical species. In addition, we compare the results of the coarser EMAC instance with the finer resolved COSMO/MESSy instance to investigate the potential benefits of the increased resolution.

The evaluation shown here is focused on June and December 2008. Results are compared to satellite observations of tropospheric $O_3$ and $NO_2$ columns, ground-level observations of $O_3$, CO and $NO_2$, to vertical $O_3$-profiles and to aircraft in situ measurements. In Sect. 2 we highlight the most important aspects of the model system and focus on differences between EMAC and COSMO/MESSy with respect to the implementation of the chemistry related submodels. Furthermore the model set-up and the chemical boundary conditions are explained. An overview about the evaluation data is given in Sect. 3, before Sect. 4 provides the comparison of model results to these observational data. In addition, a comparison of the methane lifetime in both models is given. Finally, we discuss in detail our findings about the deviations of the MECO(n) model in comparison to the observations in Sect. 5, followed by a summary and conclusion in Sect. 6.

## 2   Model description and set-up

The MECO(n) model system benefits from a key feature in the development of EMAC: Many of the chemical processes (and also diagnostic features) described in different submodels are formulated independent of the spatial and temporal scale. Therefore, most of these submodels can be used with no or little modifications in COSMO/MESSy. Readers who are not familiar with the different MESSy submodels are referred to Appendix A, which provides a general overview of the submodels that are most important for chemical processes. More details about the submodels are available on the MESSy-website (http://www.messy-interface.org) or in various publications (e.g. Jöckel et al., 2006, 2010, 2015).

An important update of the MESSy infrastructure for the use of MECO(n), however, are the new submodels IMPORT (for importing data) and GRID (for transforming between different grids) as described by Kerkweg and Jöckel (2015). In this context the old submodels ONLEM (on-line emissions), OFFLEM (off-line emissions) and DRYDEP (dry deposition) have been revised and renamed. The new submodels ONEMIS, OFFEMIS and DDEP, respectively, provide the same process parametrisations as the old process submodels, but do not include an own data import interface any more (Kerkweg and Jöckel, 2015).

### 2.1   Computational domains and on-line coupling

EMAC is used as global driving model at a resolution of T42L31ECMWF with a time step of 720 seconds. The first COSMO/MESSy instance covers the European area with a resolution of $0.44°$ ($\approx 50$ km) and integrates with a time step of 240 seconds. The size of the inner domain (neglecting the relaxation area at the model boundaries) is comparable with the domain of the EURO-CORDEX project (http://euro-cordex.net). In contrast to the EURO-CORDEX grid the domain used here is shifted and rotated slightly more to the east. We chose this different definition to be consistent with a specific high resolution emission data set, we use for another study (published elsewhere). The second COSMO/MESSy instance covers the German area with a resolution of $0.1°$ ($\approx 12$ km) and integrates with a time step of 120 seconds. This results in a MECO(2) model cascade EMAC - COSMO(50km)/MESSy - COSMO(12km)/MESSy. The regions covered by the two COSMO/MESSy instances are shown in

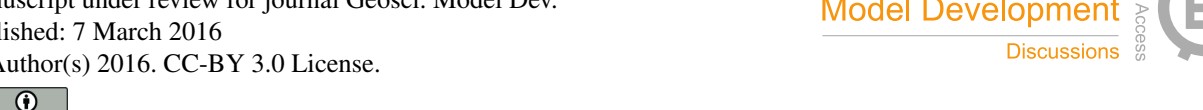

Fig. 1.

Figure 2 schematically illustrates the set-up of this MECO(2) system. In the first time step the driving model EMAC provides the necessary initial and boundary conditions for COSMO(50km)/MESSy. This COSMO(50km)/MESSy instance provides the

initial and boundary data for the COSMO(12km)/MESSy instance. For the subsequent time steps new boundary data are provided after every time step of the driving model for the finer resolved instances. Consequently, COSMO(50 km)/MESSy is receiving new boundary data every three time steps, while COSMO(12km)/MESSy is receiving updated data every two time steps.

The required transformation between the different grids are performed by the MESSy sub-submodel INT2COSMO, which

is an on-line version of the off-line preprocessing tool INT2LM provided by the German Weather Service (DWD). A detailed description is provided by Kerkweg and Jöckel (2012b). In both cases the meteorological boundary data and the boundary conditions for all chemical species (and additional diagnostic tracers) are provided.

In the MECO(n) system, model instances run in parallel within the same MPI environment. All these instances differ in their size (number of grid boxes) and the time step length. Nevertheless, these instances have to exchange data after certain model

time intervals. Thus, it is desirable, that all instances require the same wall clock time to simulate the time interval between two data exchanges to avoid idle or waiting times. Therefore, it is important to find a distribution of the MPI tasks of the participating instances on the computing system, which minimizes the waiting time between the different model instances (detailed discussion is provided by Kerkweg and Jöckel, 2012b). For the simulation set-up of this study the following distribution of MPI tasks on the 'SUPERMUC phase 1' machine at the Leibniz Supercomputing Centre (which has two 8-core processors

20  per node) is chosen: 16 tasks for EMAC, 192 tasks for COSMO(50km)/MESSy and 240 tasks for COSMO(12km)/MESSy. The optimal distribution, however, is specific for the chosen set-up and dependent on the architecture of the used computing system.

## 2.2 Simulation period and initial conditions

The simulated period ranges from 1. July 2007 until the end of 2008. The six months in 2007 are used as a spin up phase. The

25  year 2008 is evaluated. The initial conditions for EMAC and COSMO(50km)/MESSy are taken from the *RC1SD-base10a* simulation, which is described in detail by Jöckel et al. (2015). Due to the high computational costs of the COSMO(12km)/MESSy instance, this nest is only employed from 1. May 2008 until 1. September 2008.

## 2.3 Details of the model set-up

The model set-up applied here is very similar to that of the *RC1SD-base10a* simulation in the Earth System Chemistry Inte-

30  grated Modelling (ESCiMo) project described by Jöckel et al. (2015). Therefore only the most important details of the set-up and the modifications compared to the *RC1SD-base10a* set-up are summarised. An overview about the used submodels is given in Table 1. The Supplement provides full lists of the reaction mechanisms employed in MECCA and SCAV and the complete namelist set-up.





### 2.3.1 EMAC

In contrast to the *RC1SD-base10a* set-up, EMAC is applied at the resolution T42L31ECMWF here, with 31 vertical hybrid pressure levels reaching up to 10 hPa. To allow for further sensitivity studies with respect to chemical perturbations, the quasi chemistry transport model mode (QCTM-mode Deckert et al., 2011) of EMAC is used, which decouples the chemistry and the

5 dynamics. This is achieved by using climatologies for ozone (radiation), water vapour (radiation), nitric acid (heterogeneous chemistry; submodel MSBM (Multiphase Stratospheric Box Model)) and for OH, $O^1D$ and $Cl$ for methane oxidation in the stratosphere (submodel CH4). The climatologies are monthly mean values from the *RC1SD-base10a* simulation. For lightning $NO_x$ the parametrisation based on Price and Rind (1992) is chosen, which is scaled to a global nitrogen emission rate of $\approx 5\,\mathrm{Tg(N)\,a^{-1}}$ from flashes. To facilitate a comparison with observations EMAC is 'nudged' by Newtonian relaxation of

10 temperature, divergence, vorticity and the logarithm of surface pressure (Jöckel et al., 2006) towards ERA-Interim (Dee et al., 2011) reanalysis data. Sea surface temperature and sea ice coverage are prescribed as boundary conditions for the simulation set-up from this data source, too.

### 2.3.2 COSMO/MESSy

For the simulation presented here, the COSMO model in CLimate Mode (COSMO-CLM) version 5.00 is used. COSMO-

15 CLM is the community model of the German regional climate research. Besides the differences regarding the definition of the computational domain, the relaxation area and the time step the set-up of the two COSMO/MESSy instances is identical. Both instances feature 40 vertical levels ranging up to approx 24 km (20 hPa). The damping layer starts at a height of 11 km. For the time integration a Runge-Kutta scheme of third order with advection terms of fifth order is chosen. The horizontal advection is calculated using a second order Bott scheme (Bott, 1989). Most parts of the namelist set-up of COSMO are

20 identical with the COSMO-CLM set-up for the simulations within the EURO-CORDEX framework (Kotlarski et al., 2014). A detailed comparison with the CORDEX-EU set-up is part of the Supplement (Sect. S4). In COSMO no nudging of the dynamics is applied, instead the dynamics is relaxed towards EMAC at the five boundaries (four lateral boundaries and damping layer above 11 km). This means that COSMO can develop its own dynamics within the domain. As in EMAC, COSMO/MESSy is operating in a QCTM-like mode due to the prescription of the same nitric acid climatology for the MESSy submodel MSBM

as in EMAC (in fact dynamics and chemistry are decoupled as in EMAC; the overall approach differs from EMAC, therefore we use the term 'QCTM-like'). In contrast to EMAC the radiation routines of COSMO use internal climatologies. Therefore it is not possible to prescribe the same climatologies of the trace gases for the COSMO radiation routines as used in the QCTM mode for the radiation routine in EMAC. For an improved consistency between EMAC and COSMO/MESSy, it would of course be desirable to use the same climatologies for the radiation. With the current version this is not possible but it might be

implemented for future versions.



## 2.4 Chemical boundary conditions

The chemical set-up of all instances is identical, which also includes the emissions: All instances use the MACCity emissions (Granier et al., 2011) with $0.5° \times 0.5°$ grid resolution. This approach is chosen to yield a set-up, which is highly consistent from the global to the regional scale. As the same emissions are used, we are able to focus on the differences due to the change

of the basemodel (ECHAM vs. COSMO) and the increase in the resolution. For the same reason the lightning $NO_x$ emissions are calculated on-line only on the global scale. The emissions are then transformed to the grid of the regional instances. Only the emissions of soil-$NO_x$ and biogenic isoprene are on-line calculated in every instance, as the land sea mask differs between EMAC and the two COSMO/MESSy instances. Following Jöckel et al. (2006) the on-line calculated emissions of isoprene ($C_5H_8$) are scaled by a factor of 0.45 for COSMO/MESSy and 0.6 for EMAC, to be in better agreement with ground-level

observations.

## 3 Observation data

For a qualitative evaluation of the simulated tropospheric ozone and $NO_2$ columns a comparison to satellite observations is performed. For ozone the tropospheric ozone columns derived from the AURA-MLS OMI instrument (Ziemke et al., 2006) are used. The data are available as monthly mean values with a resolution of $1° \times 1.25°$.

For the calculation of the OMI tropospheric ozone columns, the definition of the tropopause according to the World Meteorological Organisation (WMO) is used. Therefore, the ozone columns of the simulation data are calculated using the on-line diagnosed tropopause height (by the submodel TROPOP) according to the WMO definition. The temperature fields employed for the calculation of the tropopause height for OMI and the simulated data are different. This can lead to differences of the diagnosed tropopause height (as discussed by Jöckel et al., 2015).

For the comparison with $NO_2$ data the satellite derived $NO_2$ measurements from the SCIAMACHY instrument (Boersma et al., 2004) with a resolution of $0.25° \times 0.25°$ are used. Similar as for the ozone columns the on-line diagnosed tropopause following the WMO definition is used as upper limit for the vertical integration of the simulation data. In both cases the averaging kernels of the measurements are not taken into account. Therefore only a qualitative comparison of the data is possible, focusing on the horizontal patterns.

For a more quantitative inter-comparison at ground-level, the simulations are compared with observations of $O_3$, $NO_2$ and CO data from the EBAS database (http://ebas.nilu.no). The choice is restricted to the data which are available for the year 2008 from the European Monitoring and Evaluation Programme (EMEP, http://www.emep.int, Tørseth et al., 2012). In addition, only those stations are selected which are within the COSMO(50km)/MESSy domain. For $O_3$ the selection is further restricted to those stations which offer observations with hourly resolution. For CO and $NO_2$ stations with daily resolution

are additionally used. The simulated vertical ozone profiles are compared with data from the world ozone database (WOUDC, http://woudc.org). All vertical profiles available in the COSMO(50km)/MESSy instance for the year 2008 are compared.

All observations are checked for a plausible range of the reported values. Finally, only data from stations with at least 75 % time coverage for the analysed period are employed. A detailed list of all station data, which are used for the evaluation, is part




of the Supplement (Sect. S5). At the latitude and longitude position of the stations the simulation data is on-line sampled with the MESSy submodel SCOUT (Jöckel et al., 2010), which samples the vertical column (of different species) at every given station. The hourly averaged SCOUT output is used for the comparison with ground-level measurements.

To allow for a fair comparison between EMAC and COSMO/MESSy always the value of the model layer which is nearest to the elevation of the station is selected. These values are referred to as 'height corrected'. Especially in mountainous terrain this is very important, as COSMO/MESSy resolves the topography much better. However, this option only works, if the station is higher than the lowest model layer. In the opposite case no extrapolation of the simulated data is performed.

For a comparison with aircraft in situ observations (CO and $O_3$) measurements from the IAGOS-CARIBIC (In-service Aircraft for a Global Observing System - Civil Aircraft for Regular Investigation of the Atmosphere Based on an Instrument
Container, Brenninkmeijer et al., 2007) project are used. For the comparison with IAGOS-CARIBIC the simulation data is sampled on-line along the flightpaths using the submodel S4D (Jöckel et al., 2010).

## 4 Evaluation

For the evaluation we focus on the results for June 2008 and December 2008, as examples for summer conditions (with strong photochemical activity) and winter conditions. First, we compare the model with results from satellite measurements of the
15 tropospheric ozone and $NO_2$ columns (Sect. 4.1). Subsequently, the differences between the simulation data and the ground-level observations (Sect. 4.2), the vertical ozone profiles (Sect. 4.3) and aircraft in situ observations (Sect. 4.4) are investigated. At the end the simulated methane lifetimes are analysed in view of the tropospheric oxidation capacity (Sect. 4.5).

The simulated meteorology of EMAC and the two COSMO/MESSy instances is also compared to ERA-Interim data (Dee et al., 2011) and to the vertical temperature profiles from the ozone sonde data, which are used in Sect. 4.3. We do not focus
on the discussion of meteorology in this study, as the meteorological evaluation of MECO(n) has already been performed by Hofmann et al. (2012), but rather provide the main results:

In general a cold bias exists throughout the year in both COSMO/MESSy domains in the troposphere. EMAC shows only a little or no cold bias in the lower troposphere. A strong cold bias is present in EMAC in the upper troposphere, which is not that prominent in COSMO/MESSy. The cold bias of COSMO/MESSy results in a slightly enhanced positive bias of the mean sea
level pressure compared to EMAC. For the 10 meter wind speed EMAC shows a small negative bias, while COSMO/MESSy mainly shows a positive bias near the coastlines. The corresponding figures are part of the Supplement (Sect. S2 and Sect. S3).

### 4.1 Comparison with satellite observations

Figure 3 shows the ozone columns of OMI (left), EMAC (centre) and COSMO(50km)/MESSy (right) for June 2008. Please note, that the OMI data are scaled for a better comparability. For the reasons discussed is Sect. 3, it is not possible to derive the
30 magnitude of the bias for ozone from this scaling factor. The bias for ozone is quantified in the following sections. The overall patterns of all three ozone columns look very similar with a strong north-south gradient. Investigating into more detail, some differences are apparent. The maximum over the Mediterranean sea is underestimated in COSMO(50km)/MESSy. EMAC



simulates a higher extend of this maximum, which better corresponds to the satellite measurements. The low values over the Alps or the Atlas mountains in Morocco found in the OMI data are well reproduced by COSMO(50km)/MESSy. Also the higher ozone values in South-West France, which are present in the OMI data, are better reproduced by COSMO(50km)/MESSy in comparison to EMAC. Over Poland, the Baltic Sea and East Germany COSMO(50km)/MESSy shows higher values compared to EMAC and OMI. For December 2008 the OMI data are very noisy over Europe, therefore we are do not present a comparison for this month.

Figure 4a shows the monthly averaged tropospheric $NO_2$ columns for June 2008. In general COSMO(50km)/MESSy captures the hotspot regions much better than EMAC due to the higher resolution. Some examples are the Po basin, Paris, Madrid, Moscow, the Eastern Ukraine and the coastal regions of the Middle East. Striking is the overestimation of $NO_2$ in COSMO(50km)/MESSy in South-East Europe. Furthermore, some other hotspots are overestimated by the MACCity emission database e.g. the region around Helsinki or the harbour area around Marseilles.

For December 2008 (Fig. 4b) we see overall a similar picture. Due to the coarse resolution of EMAC the emissions are spread over large gridboxes, which make it hard to resolve individual hotspots. A good example is the Po basin region, which does not exist in EMAC. In COSMO(50km)/MESSy this hotspot is underestimated. The $NO_2$ columns simulated by COSMO(50km)/MESSy over the Atlantic sea between Spain and England are overestimated, possibly due to overestimated ship emissions in this area. Additionally COSMO(50km)/MESSy overestimates most hotspots like in England and Germany. This overestimation indicates that the $NO_x$ emissions are too high in this region or that too much NO is converted to $NO_2$ by reaction with $O_3$ or $HO_2$.

## 4.2 Comparison with ground-level measurements

Figure 5a shows the monthly averaged ozone concentrations of EMAC (left) and COSMO(50km)/MESSy (right) for June 2008. The ozone concentrations of the lowest model layer are displayed as coloured contours. The coloured symbols indicate the positions of the observations and compare simulated and observed ozone concentrations at the measurement sites.

In comparison to EMAC, COSMO(50km)/MESSy shows a better agreement with the observations over Germany, France and Spain. Comparing the monthly averaged values at all measurement sites, both models show an overall positive ozone bias with a normalized mean bias error (MBE) of around 16% for EMAC and 20% for COSMO(50km)/MESSy (see Table 2). Compared to EMAC this positive bias in COSMO(50km)/MESSy is more pronounced over North-East Europe, than over Central Europe. This bias is further discussed in Sect. 5.

In general this bias is slightly lower than the MBE of around 30% (1.875° x 1.25° resolution) and 33% (0.56° x 0.375° resolution) found by Stock et al. (2014) over Europe using the UKCA model. As they calculated the MBE for July 2005 we additionally calculated the MBE for July 2008, which is 18% for EMAC and 17% for COSMO(50km)/MESSy. In comparison, Knote et al. (2011) found negative values of the MBE between -3% and -15% for summer conditions in June 2006 using the COSMO-ART model.

Figure 5b displays the simulated ozone concentrations for June 2008 zooming in on Germany. While the values for COSMO(50km)/MES are shown on the left, the values for COSMO(12km)/MESSy are shown on the right. In general, the ground-level ozone distri-



bution is very similar, though much more details are revealed by the enhanced resolution of the second instance. As the same $0.5° \times 0.5°$ emission database is used, the differences are due to the more realistic topography (e.g. the Rhine valley or the Eifel region). Compared to the measurements the root-mean-square error (RMSE) slightly decreases with finer resolution, from $15\,\mu g\,m^{-3}$ in EMAC to $12\,\mu g\,m^{-3}$ in COSMO(50km)/MESSy and to $11\,\mu g\,m^{-3}$ in COSMO(12km)/MESSy. The MBE is de-

creasing from 10% in EMAC to 4% in COSMO(50km)/MESSy and increases again to 7% in COSMO(12km)/MESSy (Table 3). While the benefit of the increased resolution (detected in a decreased RMSE and MBE) compared to EMAC is obvious, it is important to note again, that both COSMO/MESSy instances are using the same emissions with $0.5° \times 0.5°$ resolution. A detailed investigation of the effect of the finer resolution of COSMO(12km)/MESSy compared to COSMO(50km)/MESSy is beyond the scope of this study and requires a different experimental set-up, with adequately resolved emissions and an inter com-

parison with a dense local measurement network like AirBase (European Air quality dataBase, http://airbase.eionet.europa.eu).

The ground-level ozone concentrations in COSMO(50km)/MESSy for December 2008 (Fig. 5c) show more details compared to EMAC. Examples are the higher values in the mountainous areas (Alps, Pyrenees) and lower values in hotspot regions like the Po valley or around Paris. Comparing the height corrected values at the mountain stations EMAC and COSMO/MESSy show comparable results. The reason for this difference between the ground-level concentrations and the 'height corrected'

concentrations are the better resolved topography in COSMO/MESSy compared to EMAC. Apparent is also the enhanced positive bias of COSMO/MESSy over Middle and North-Eastern Europe: the MBE is around 20% in EMAC and 28% in COSMO(50km)/MESSy. In comparison to this, Knote et al. (2011) found a negative bias of an approximately similar amplitude (22%) for winter conditions in COSMO-ART.

As already seen in the comparison with the SCIAMACHY $NO_2$ columns, the increased resolution of COSMO(50km)/MESSy

shows the largest benefit when comparing ground-level $NO_2$ concentrations of EMAC and COSMO(50km)/MESSy with observations. The monthly mean nitrogen dioxide concentrations for June 2008 are shown in Fig. 6a. Comparing the simulated concentrations from EMAC (left) and COSMO(50km)/MESSy (right) to measurements, the highly variable regional distribution, with higher concentrations near the hotspots and lower concentrations in the remote areas, is better represented by COSMO(50km)/MESSy. The RMSE (Table 2) of EMAC and COSMO(50km)/MESSy is similar ($\approx 1\,\mu g(N)\,m^{-3}$). Accord-

ing to the MBE, both models show a negative bias. This bias is $\approx 16\%$ larger in COSMO(50km)/MESSy than in EMAC. However, this quantity does only compare the average over all stations; positive and negative biases at different stations cancel out.

For the stations located the COSMO(12km)/MESSy domain similar results are found. The RMSE between EMAC and the two COSMO/MESSy instances is similar, while the negative bias of the MBE is larger in both COSMO/MESSy instances

compared to EMAC. The corresponding Fig. displaying the ground-level concentrations is part of the Supplement (Sect. S1.5).

A similar picture as for June 2008 is found for December 2008 (Fig. 6b). Comparing first the ground-level concentrations between EMAC and COSMO(50km)/MESSy a higher contrast between remote areas and hotspot regions is present in COSMO(50km)/MESSy. In comparison to the measurements the strong contrast between hotspot and remote regions is simulated better by COSMO(50km)/MESSy than by EMAC (e.g. south of Spain, Norway). As for June, both models show a

negative MBE (-42% for EMAC, -46% for COSMO(50km)/MESSy), the RMSE is similar ($3\,\mu g(N)\,m^{-3}$).



Despite the better representation of hotspots in COSMO(50km)/MESSy some measured concentrations are underestimated in COSMO(50km)/MESSy (and EMAC). These hotspots may not be covered by the emission database, or local effects, which cannot be resolved by or are missing in the model, play an important role.

The MBE for COSMO(50km)/MESSy is -38% (June) and -48% (December) which is within similar ranges as reported by Knote et al. (2011) for $NO_2$ using COSMO-ART. However, they report a positive, not a negative bias. The difference of the sign might be explained by the different emission data sets, as they used the emission data set provided by TNO (Netherlands) with an hourly time curve (Kuenen, 2011), while the MACCity data set with a constant emission flux for the whole month is used here.

Simulated ground-level CO mixing ratios in June 2008 (Fig. 7a) and December 2008 (Fig. 7b) show a negative bias in EMAC and in COSMO(50km)/MESSy. Again the larger regional variation of the ground-level mixing ratio with lower values over the Alps, as well as the larger values over the largely polluted Po valley, can be resolved much better by COSMO(50km)/MESSy. Comparing the height corrected values the MBE is around -20% for EMAC (independent of season) and between -25% (December) and -28% (June) for COSMO(50km)/MESSy (Table 2). The difference of the RMSE between EMAC and COSMO is similar for June (around 4 $nmol\ mol^{-1}$) and in December (6 $nmol\ mol^{-1}$).

Additional comparisons of simulated ground-level concentrations with observation of isoprene ($C_5H_8$) and nitric acid ($HNO_3$) are part of the Supplement (Sect. S1.6, S1.7). Both species are simulated well in COSMO(50km)/MESSy compared to the observations. Especially for $C_5H_8$ the benefit of the increased resolution is obvious, because the larger spatial variability of the observations is captured much better by COSMO(50km)/MESSy than by EMAC.

### 4.2.1 Taylor diagrams

For a more quantitative comparison Taylor diagrams (Taylor, 2001) are calculated. These diagrams combine the (normalised) standard deviation and the correlation between the observed and the simulated time series (based on hourly averaged model output/observations). The bias in percent between the simulated and observed ozone concentration is displayed by the size of the symbols. Again, only the height corrected values are used, which improve the results of EMAC considerably. The Taylor diagrams for the uncorrected cases are part of the Supplement (Sect. 1.4).

The corresponding Taylor diagrams for June and December 2008 are shown in Fig. 8. In addition to the individual stations for EMAC and COSMO/MESSy also the mean over all stations for every model is depicted. The symbols below the horizontal axis indicate stations with a correlation or standard deviation out of the range displayed in the corresponding diagrams.

For June 2008 both models underestimate the variability of the observations. The mean values for the normalised standard deviation are larger in EMAC (0.74) compared to COSMO(50km)/MESSy (0.65). The same is true for the correlation coefficient which is 0.48 for EMAC and 0.34 for COSMO(50km)/MESSy. In general the results at different stations in both models are similarly scattered. The biases of EMAC (17%) and COSMO(50km)/MESSy (22%) are positive.

For December 2008, both models show a better agreement with the observed normalised standard deviations. For EMAC the mean normalised standard deviation increases to 0.97, while the normalised standard deviation for COSMO(50km)/MESSy in-



creases to 0.78. The mean correlation coefficients for both models decrease to 0.45 for EMAC and 0.38 for COSMO(50km)/MESSy respectively. As for June the results at different stations in EMAC and COSMO(50km)/MESSy are similarly scattered.

We also calculate the Taylor diagrams for the entire year 2008 (Fig. 9). In this case the correlation is higher than 0.50 (0.63 for EMAC and 0.55 for COSMO(50km)/MESSy). The standard deviation is 0.84 for EMAC and 0.73 for COSMO(50km)/MESSy. This indicates, that the amplitude of the annual cycle is underestimated by both models, while the general shape is well simulated by both models. Some exemplarily figures comparing the annual cycle of EMAC and COSMO(50km)/MESSy with the observation are part of the Supplement (Sect. 1.3)

### 4.2.2 Diurnal Cycles

To compare the diurnal cycle at the different stations, we calculate an average diurnal cycle for all non-mountain stations (stations with an elevation lower than $800$ m) and all mountain stations. Again, the height corrected model data is used. For a more quantitative analysis we split these averaged diurnal cycles into mean values and the amplitudes. For this we calculate first the monthly averaged diurnal cycle at every station. From this cycle, the mean value is calculated, which is subtracted from the diurnal cycle to get the amplitude of the diurnal cycle. These values are averaged in a second step over all non-mountain/mountain stations.

Figure 10a shows the averaged amplitude of the diurnal cycle of the non-mountain stations for June 2008, the corresponding mean values are listed in Table 4. Comparing the mean values of EMAC and COSMO(50km)/MESSy the positive ozone bias is apparent, however the differences are within one standard deviation of the observations. The amplitude, however is underestimated in COSMO(50km)/MESSy. While the amplitude of the observations is in the range of $\pm 18$ µg m$^{-3}$, COSMO(50km)/MESSy simulates an amplitude of only $\pm 5$ µg m$^{-3}$. EMAC exhibits an amplitude of $\approx \pm 12$ µg m$^{-3}$. Comparing not the amplitude, but the complete diurnal cycle (not shown), both EMAC and COSMO(50km)/MESSy simulate an identical noon peak of $\approx 100$ µg m$^{-3}$ (the observations show a peak of $\approx 93$ µg m$^{-3}$). In fact COSMO(50km)/MESSy underestimates the decrease of ozone during night ( which is mainly due to chemical destruction and dry deposition). This issue is discussed in detail in Sect. 5.

For the mountain stations in June 2008 COSMO(50km)/MESSy simulates mean values, which are comparable with the observations, while EMAC shows a positive ozone bias ($\approx 7$ µg m$^{-3}$). However, the small amplitude of the observed diurnal cycle ($\pm 4$ µg m$^{-3}$) is underestimated by both models, which show hardly any amplitude.

Figure 11 displays the averaged amplitude of the diurnal cycle for the subset of stations, which are located in both COSMO/MESSy instances. The corresponding mean values are listed in Table 5. Overall, the results are similar as for all stations. For the non-mountain stations EMAC and the two COSMO/MESSy instances underestimate the observed amplitude of the diurnal cycle ($\approx \pm 19$ µg m$^{-3}$). Especially, the two COSMO/MESSy instances reach smaller ($\approx \pm 5$ µg m$^{-3}$) values compared to EMAC ($\pm 12$ µg m$^{-3}$). The absolute values of the observed noon peak (not shown) are well simulated by both COSMO/MESSy instances ($\approx 95$ µg m$^{-3}$) and overestimated by EMAC ($\approx 102$ µg m$^{-3}$). Again, the loss over night is underestimated in COSMO(50km)/MESSy. For the mountain stations EMAC and the two COSMO/MESSy instances do not reproduce the small



amplitude ($\approx \pm 10\ \mu g\ m^{-3}$). COSMO(12km)/MESSy shows the largest amplitude ($\approx \pm 2\ \mu g\ m^{-3}$). The mean values have a negative bias for both COSMO/MESSy instances ($\approx -5\ \mu g\ m^{-3}$) and a positive bias for EMAC ($\approx 5\ \mu g\ m^{-3}$).

For the mountain stations in December 2008, both models in general simulate a similar amplitude compared to the observations (Fig. 12a). However the (small) noon peak is underestimated, yet all differences are within one standard deviation of the observations. The mean values, show a positive bias of $\approx 19\ \mu g\ m^{-3}$ for EMAC and $\approx 29\ \mu g\ m^{-3}$ for COSMO(50km)/MESSy (Table 4).

This bias for ozone exists also at the mountain stations, but smaller in magnitude ($8\ \mu g\ m^{-3}$ for EMAC and $13\ \mu g\ m^{-3}$ for COSMO(50km)/MESSy); the absence of a diurnal cycle is represented by both models (Fig. 12b).

### 4.3 Vertical ozone profiles

In order to check, if the vertical distribution of ozone is well simulated, we compare the simulation results with ozone sonde data. For this, the ozone sonde data are transformed to a fixed pressure grid. The ozone sonde data are not continuous measurements in time, but represent distinct points in time (and space). To simplify the comparison with the simulated data all measurements within one month are averaged, without any weighting of the individual measurements. From the simulations, we use the hourly averaged model output at the location of every station, which are averaged over the month. Therefore the simulated and observed data is co-located in space, but not necessarily in time.

Exemplarily the ozone profiles of the observations and from the simulation data at Hohenpeissenberg (Fig. 13) are displayed. For June 2008 additionally also the vertical profiles for COSMO(12km)/MESSy are shown. Profiles at more stations can be found in the Supplement (Sect. 1.8). The vertical ozone distribution is captured well by EMAC and all COSMO/MESSy instances. For most profiles, the mean of the simulated ozone mixing ratios lies within one standard derivation of the mean from the observations. However, in the boundary layer we note a positive bias of COSMO/MESSy at most stations. This bias is in line with the results already presented above. The large variability of the observations in the upper troposphere/lower stratosphere (UTLS) area is captured much better by COSMO/MESSy than by EMAC, as COSMO/MESSy resolves intrusion of stratospheric air into the troposphere better. However, while comparing the variability, it is again important to note, that the number of data points of the observations is much lower than for the simulated data. The results of COSMO(12km)/MESSy (Fig. 13) are very similar to COSMO(50km)/MESSy, but the variability is slightly larger due to the finer horizontal resolution.

Despite the good representation of the measured ozone mixing ratios in the free troposphere ozone is overestimated within the planetary boundary layer (PBL) at most stations, which is more pronounced in COSMO/MESSy than in EMAC. For some stations (e.g. Payerne, Legionowo) even only a small or no gradient of the mixing ratio within the PBL is simulated by COSMO/MESSy. This problem is discussed in detail in Sect. 5.

In addition, we calculate the RMSE between the monthly mean data of the observations and the simulations. For this, the observations are transformed on the vertical grid of the respective simulation. The RMSE for all profiles in June 2008 is shown in Fig. 14a. In general the RMSEs of EMAC and COSMO(50km)/MESSy look very similar. From the bottom up to roughly 800 hPa the RMSEs are between 0 - 20 $nmol\ mol^{-1}$. From 800 hPa to 600 hPa the RMSE increase to 5 - 25 $nmol\ mol^{-1}$.



At 600 hPa they drop back to 0 - 20 nmol mol$^{-1}$. In the UTLS area the variability of the RMSE is increasing again. In this area the variability and the absolute ozone values are very large.

In December 2008 (Fig. 14b) too high values within the PBL in COSMO(50km)/MESSy show up by higher values of the RMSE (up to 25 nmol mol$^{-1}$), while EMAC exhibits a maximum RMSE of 15 nmol mol$^{-1}$. At roughly 800 hPa both models
show a decreased RMSE of $\approx 10$ nmol mol$^{-1}$ at maximum, before the spread of the RMSE is again increasing in the UTLS area.

### 4.4 In-situ observations

We here compare exemplarily the simulation results of EMAC and COSMO(50km)/MESSy with measurements of the IAGOS-CARIBIC flight 240 from Frankfurt (Germany) to Chennai (India) and the flight 243 from Denver (USA) to Frankfurt (both
July 2008). The flight was sampled in EMAC and in COSMO(50km)/MESSy using the MESSy submodel S4D (Jöckel et al., 2010), which on-line samples the model data along the flight path with model time step resolution. For a better comparison between simulated and measured data the measurements are aggregated on the same time step as the model output (720 seconds for EMAC and 240 seconds for COSMO(50km)/MESSy). Ozone and carbon monoxide mixing rations from the simulation and the measurements are compared in Fig. 15. For the simulation data additionally the potential vorticity (PV) is displayed. In
general both models underestimate carbon monoxide and overestimate ozone in the troposphere. This is in line with the findings of the previous sections. However the intrusion of stratospheric air at the beginning of the flight 240 is captured much better by COSMO(50km)/MESSy. This is visible from the high values of the ozone mixing ratios, where the observed magnitude is nearly perfectly reproduced by COSMO(50km)/MESSy. Flight 243 resides in stratospheric air masses most of the time. Here the carbon monoxide mixing ratios are well simulated by both models. However, the huge fluctuations of the ozone mixing
ratios along the flight track are not captured by the models. To achieve this maybe a higher vertical resolution is necessary to account for the steep vertical gradients in the UTLS area. Also note, that parts of the flight may already be within the upper damping zone (starting at 11 km) of COSMO(50km)/MESSy. For future comparisons the use of a grid with a higher vertical extent in COSMO/MESSy (e.g., Eckstein et al., 2015) are envisaged.

### 4.5 Tropospheric oxidation capacity

To compare the oxidation capacity of the troposphere between EMAC and the two COSMO/MESSy instances the lifetime of methane against OH ($\tau_{CH_4+OH}$) is calculated according to Jöckel et al. (2006) as

$$\tau_{CH_4+OH} = \frac{\sum_{b,t} M_{CH_4}^b(t)}{\sum_{b,t} \kappa_{CH_4+OH}^b(t) \cdot c_{air}^b(t) \cdot OH^b(t) \cdot M_{CH_4}^b(t)} \quad (1)$$

with $M_{CH_4}^b(t)$ the mass of CH$_4$ in every gridbox (b) at a respective time step (t), $\kappa_{CH_4+OH}^b(t)$ the reaction coefficient of the reaction CH$_4$ + OH (which depends on the temperature), $c_{air}^b(t)$ the concentration of air and OH$^b(t)$ the mole fraction of OH.
Usually, the lifetime of methane is calculated in global models. In this case the methane lifetime can be calculated at every time step. As we calculate the lifetime only for a fraction of the globe, it is important to sum the numerator and denominator



first over all time steps of a certain period (> 1 day) before the calculation of $\tau$. The reason for this is, that during night OH is virtually absent and the denominator becomes arbitrary small.

We calculate the methane lifetime for three different vertical layers of the atmosphere: From the ground to 850 hPa, from 850 hPa to 500 hPa and finally from 500 to 200 hPa. For this we sum up all grid boxes within the respective area.

First, we compare $\tau$ for the German region, which is covered by EMAC, COSMO(50km)/MESSy and COSMO(12km)/MESSy (Table 6). For the layer from the bottom up to 850 hPa EMAC calculates the shortest average lifetime (2.7 a), which is due to a larger OH mass (60 kg). In the COSMO(12km)/MESSy instance the lifetime is considerably shorter (2.9 a) than in COSMO(50km)/MESSy (3.4 a), as more OH is present in the finer resolved instance. In the second vertical layer (850 - 500 hPa), both COSMO/MESSy instances show comparable results (3.5 a). The $CH_4$ mass is smaller compared to EMAC, while

the OH mass is larger, which leads to a shorter average $CH_4$ lifetime in both COSMO/MESSy instances compared to EMAC. For the highest vertical layer (500 - 200 hPa) all instances show comparable OH masses, the lifetime of methane, however, is longer for EMAC (12.4 a) compared to COSMO(50km)/MESSy (11.3 a) and COSMO(12km)/MESSy (11.2 a). This difference is mainly caused the lower temperatures in EMAC in this vertical layer.

The methane lifetimes in the European domain (Table 7) show similar results as over Germany. In the lowest vertical layer

EMAC simulates a shorter methane lifetime (mainly due to more OH). In the second vertical layer both models simulate very similar methane lifetimes, while the lifetime in the upper layer is again larger in COSMO(50km)/MESSy. The shorter lifetime in EMAC compared to COSMO/MESSy is due to more OH and a higher temperature in EMAC.

## 5 Discussion on deviations from observations

By comparing the COSMO/MESSy results with observations in the previous section, we find some remarkable deviations.

First of all the simulated ground-level mixing ratios of carbon monoxide are too low, while the ozone concentrations are too high. In particular the North-East European area is affected by too high ground-level ozone concentrations during April (not shown) to June. In addition, not only the monthly mean ground-level concentrations of ozone are too high, also the amplitude of the diurnal cycle is underestimated showing too large values in COSMO(50km)/MESSy at night.

To investigate the influence of the cold-bias of COSMO/MESSy (which is a known problem of COSMO-CLM during winter

e.g., Kotlarski et al., 2014), we conduct a short sensitivity study with a modified temperature field of COSMO(50km)/MESSy for the calculation of the reaction kinetics in the submodel MECCA (see Appendix A). For this the temperature field of EMAC is transformed using INT2COSMO to COSMO(50km)/MESSy. This transformed temperature field is then used within MECCA. All other dynamical and chemical processes (like the on-line calculation of emissions) use the original temperature field of COSMO(50km)/MESSy. Resulting area averaged ground-level concentrations for a small subset of all chemical species

over Europe (defined as a box from 5°W - 20°E, 20°N - 55°N) are summarised in Table 8.

Comparing first the area averaged concentrations between EMAC and COSMO(50km)/MESSy we see for all species, except ozone, a positive difference which means higher values in EMAC compared to COSMO(50km)/MESSy. This includes short lived tracers like OH or $NO_3$ and longer lived tracers like bromoform ($CHBr_3$) and CO. Comparing further the re-





sults between COSMO(50km)/MESSy and COSMO(50km)/MESSy$_{T^*}$ (with the changed temperature field) we see that the concentrations of most short lived species (like OH, NO$_3$ or HCHO) increase. This difference is due to the temperature dependence of most reaction rates. The magnitude of this increase can, however, not fully explain the observed difference between COSMO(50km)/MESSy and EMAC, but it is one important contributor to the difference of the short lived tracers between

EMAC and COSMO(50km)/MESSy.

The differences of longer lived species like ozone, carbon monoxide or bromoform can not be explained by the temperature difference. For further analysis, we compare vertical profiles of $^{222}$Radon (using the MESSy submodel DRADON, Jöckel et al., 2010) in COSMO(50km)/MESSy and EMAC. This submodel emits $^{222}$Radon as purely diagnostic species on all land surfaces not covered by ice or snow. The emission rate is 10000 $\mathrm{atoms\,m^{-2}\,s^{-1}}$ and the only sink in the atmosphere is

radioactive decay with a half-life of 3.8 days.

The vertical profiles of $^{222}$Radon (not shown) show smaller concentrations in the PBL in COSMO/MESSy than in EMAC, even though the sources are identical. This difference can only be explained by a stronger vertical mixing (vertical diffusion) within the PBL in COSMO/MESSy compared to EMAC. This stronger mixing explains also the differences for the longer lived trace gases like ozone, carbon monoxide or bromoform. For CO and bromoform the high concentrations near the surface

are more quickly reduced through upward transport in COSMO/MESSy than in EMAC. The concentration of ozone increases with height, meaning that the lower values at the surface are faster mixed with air containing more ozone. This is in agreement with the vertical ozone profiles of COSMO(50km)/MESSy (see Sect. 4.3) showing too large ozone mixing ratios in the PBL.

In addition to this stronger mixing there is yet another cause for the too high ozone concentration in COSMO/MESSy over North-East Europe: COSMO/MESSy uses different soil types in some areas over North-Eastern Europe. This affects for exam-

ple the stomata resistance determined by the different base models, which subsequently affects the dry deposition velocities. This leads to a reduced dry deposition velocity over parts of North-Eastern Europe in COSMO/MESSy compared to EMAC (additional figures are part of the Supplement in Sect 1.1). Moreover Stock et al. (2014) found higher ground-level concentrations of ozone over North-Eastern Europe, when increasing the resolution of their simulations. As they are using the same MACCity emissions as we do, we speculate that the too large ground-level mixing ratios of ozone might also be influenced

by too large emissions of ozone precursors in this area. As the ozone chemistry is strongly non-linear even a small amount of higher NO$_x$ emissions would lead to an increased ozone production in the NO$_x$-limited regime.

The underestimation of the amplitude of the diurnal cycle in COSMO/MESSy has several reasons. The most important difference is the dynamics of the PBL. The diurnal cycle of the PBL is more pronounced in EMAC compared to COSMO(50km)/MESSy,

showing higher values around noon and smaller values during night (Fig. 16).

The lower height of the PBL in EMAC during night leads to a much smaller 'reservoir' from which ozone can be deposited or chemically destroyed (e.g. via reaction with NO). Nevertheless the amount of ozone which is removed by dry deposition depends on the concentration of ozone, the concentration in this 'smaller reservoir' is faster reduced during night in EMAC compared to COSMO(50km)/MESSy. This leads in general to a more efficient destruction of ground-level ozone during night,




when no photochemically production of ozone takes place. In addition the more efficient vertical diffusion in COSMO/MESSy (as discussed above) leads to more efficient downward transport of air with higher ozone concentration.

This is intensified by two additional differences between EMAC and COSMO/MESSy leading to a more pronounced diurnal cycle in EMAC. First of all, the dry deposition velocities during noon are comparable between COSMO/MESSy and EMAC.
During night this changes and EMAC simulates a slightly larger dry deposition velocity as COSMO/MESSy. In addition, the net ozone production in the lowermost model layer (production - loss) is more negative during night in EMAC compared to COSMO/MESSy.

To investigate, if we can improve the vertical ozone profiles and the amplitude of the diurnal cycle of ozone in COSMO/MESSy by changes to the COSMO set-up, we conducted further sensitivity studies. The main aim of these studies was to investigate
the effect of changing parameters affecting vertical mixing (diffusion).

Focusing on the vertical ozone profiles in comparison to ozone sonde observations and the amplitude of the diurnal cycle of ozone, none of these simulations shows substantial improvements compared to the observations.

One simulation, however, slightly improves the amplitude of the diurnal cycle and shows a decreased cold-bias. Compared to the reference set-up, the minimum diffusion coefficient for temperature (tkhmin = 0.1) and momentum (tkmmin = 0.1) is
decreased. Further the factor for diffusion of turbulent kinetic energy (TKE, c_diff=0.05), the length scale for sub-scale surface pattern (pat_len=100) and the maximal turbulent length scale (tur_len=150) are decreased. In addition, also the explicit corrections of implicitly calculated turbulent heat and moisture fluxes due to effects from subgrid-scale condensation is switched of (lexpcor = false, which is also set to false for COSMO-DE and COSMO-EU at the DWD or in the CORDEX-EU set-up). We recommend these settings for further simulations using COSMO/MESSy at least over Europe and with a resolution comparable
to the simulations performed here. Using an increased resolution or a domain in different regions of the world might require other parameters.

To improve the results with respect to the too small amplitude of the diurnal cycle of the PBL and the too strong mixing within the PBL further model developments are necessary. For example, the turbulence scheme and thus the vertical diffusion parametrisation were recently further developed for the ICON model (pers. communication, M. Raschendorfer, DWD). These
developments become available in the COSMO model from version 5.3 on. Further testing of the additional options available within this newer COSMO version are planned as soon as these are available. In this context, a detailed comparison with observed diurnal cycles for temperature and relative humidity between COSMO/MESSy and observations are required.

Furthermore, it is well known, that the soil moisture has an important influence on the boundary layer dynamics. Therefore, a better initialisation of the soil moisture could very well yield an improved diurnal cycle and more realistic vertical profiles.
In future, additional tests with a nudging of the mean temperature in EMAC (as done in some of the simulations described by Jöckel et al., 2015) would be interesting, to test whether the cold bias in the upper troposphere can be reduced.



## 6 Summary and Conclusion

For the first time we perform model simulations using complex tropospheric and stratospheric chemistry with the newly developed model system MECO(n). MECO(n) features an on-line coupling between the global chemistry-climate model EMAC and the regional chemistry-climate model COSMO/MESSy. The main purpose of the simulations is the evaluation of MECO(n) with respect to gas-phase chemistry. This evaluation is a prerequisite for further studies focusing on the analysis of atmospheric chemistry. Therefore, we perform a simulation covering the period from July 2007 - December 2008, from which we compare the results for June and December 2008 to observations. We use a MECO(2) set-up with one regional instance covering Europe $(0.44°)$ and a second instance covering Germany $(0.1°)$. Because of the high computational demands the finer nest was applied only during the summer period of 2008. The chemical boundary conditions of EMAC and the two COSMO/MESSy instances were as consistent as possible. This means, that we use the same emission data set with a resolution of $0.5° \times 0.5°$ for all instances and the same lightning $NO_x$ emissions as calculated by EMAC in all instances. This set-up allows us to focus on the difference due to the changes of the base model (ECHAM vs. COSMO) and the increased resolution.

We focus on the evaluation of ozone, carbon monoxide and nitrogen dioxide and compare the simulated values with satellite observations, in situ ground-level data, vertical profiles and aircraft in situ measurements. This comparison shows, that the increased resolution of COSMO/MESSy allows for a more detailed representation of the hotspot regions. Especially, the spatial representation of highly variable trace gases like nitrogen dioxide are improved. The annual cycles of the investigated trace gases are represented well by COSMO/MESSy and by EMAC. Especially for the German area we found a better agreement with observations using COSMO/MESSy instead of EMAC. The same is true for the representation of ozone at mountain stations.

COSMO/MESSy shows a positive bias for ozone and a negative bias for nitrogen dioxide. The magnitude of the bias is in the same range as for that of comparable model systems. In addition also a negative bias for carbon monoxide is apparent. The vertical profiles of COSMO/MESSy are in agreement with observations from ozone sonde data within the free troposphere, showing a RMSE between 0 - 20 $nmol\,mol^{-1}$. Especially the large variability in the UTLS region is captured much better by COSMO/MESSy as by the coarser resolved EMAC model. This shows the high potential of MECO(n) for the preparation and wrap-up of aircraft measurement campaigns helping to interpret the measurements.

The diurnal cycle of ozone is not as good represented in COSMO/MESSy as in EMAC. The main reason for this are differences in the dynamics of the models. Especially the amplitude of the diurnal cycle of the PBL is smaller in COSMO/MESSy compared to EMAC. The comparison of the vertical profiles from COSMO/MESSy to observations shows, that the profiles within the PBL at some stations in COSMO/MESSy are to steep. The COSMO/MESSy profiles are also steeper compared to EMAC, explaining the increased positive ozone and negative carbon monoxide bias in COSMO/MESSy. In order to overcome these problems further model improvements are necessary, e.g. the improvement of the PBL turbulence scheme.

It is also important to note, that the potential of the increased resolution (especially for the finest instance) is not fully exploited in the simulation presented here, as a coarse emission data set is used in all instances. Usage of coarse emission data sets can lead to deterioration of the results on finer scales, as the emissions are already blurred out due to the coarse



resolution of the emission data and small peaks on a scale smaller than the emission data can not be resolved. A finer resolved emission data set is expected to reveal much more benefits of the increased resolution. This, however, is not the intention of the simulation presented here. The purpose of this study is a first evaluation of the MECO(n) model system with respect to tropospheric chemistry. This evaluation is an important step in the model development. We show that both models have

strengths and weaknesses. Even with coarse emission data COSMO/MESSy shows its strength in particular in the comparison with in situ aircraft observations. Besides further model improvements, the next step will be a detailed evaluation using high resolution emissions and comparison with regional observation networks.

## 7    Code availability

The Modular Earth Submodel System (MESSy) is continuously further developed and applied by a consortium of institutions.

The usage of MESSy and access to the source code is licensed to all affiliates of institutions which are members of the MESSy Consortium. Institutions can become a member of the MESSy Consortium by signing the MESSy Memorandum of Understanding. The legacy model ECHAM5 is licensed by the Max-Planck Institute for Meteorology in Hamburg (Germany). The COSMO code is available under two different licenses: Either an individual user license granted by the CLM-Community or by an institutional license granted by the German Weather Service (DWD). More information can be found on the MESSy

Consortium Website (http://www.messy-interface.org).

## Appendix A:   Description of gas-phase chemistry related submodels

Due to the modular MESSy infrastructure, we can use most of the submodels of the MESSy framework simultaneously in EMAC and COSMO/MESSy. This is especially the case for all submodels, which are important for the calculation of atmospheric chemistry. Below we provide a short overview of the submodels which are most important for the calculation of

atmospheric chemistry processes. We restrict this overview to the submodels (with the exception of MECCA), where differences between EMAC and COSMO/MESSy exist.

In the beginning we would like to highlight one general important difference between COSMO/MESSy and EMAC with respect to the submodels DDEP (dry deposition), OFFEMIS (off-line emissions) and ONEMIS (on-line emissions). In general these submodels have two options to handle the deposition and emissions: the tracer tendency in the respective model box can

be directly changed or a lower boundary condition for the vertical flux can be calculated. In the latter case, the emission is treated by the vertical diffusion operator (VDIFF, more details can be found in Kerkweg et al., 2006b). In general both options would be available for use in COSMO/MESSy. However, as using the lower boundary flux can lead to problems in closing the budgets of the trace species in COSMO/MESSy only the option to change the tracer tendencies directly has been implemented, so far.

– DDEP

     The submodel DDEP handles the dry deposition of trace gases and aerosols. Following the approach of Wesely (1989)



the dry deposition velocities of ozone and sulphur dioxide are calculated explicitly, as these dry deposition velocities are relatively well known. The velocities of the other trace gases are calculated in relation to the velocities for ozone and sulphur dioxide depending on their solubility and reactivity. The only exceptions are nitrogen oxide, nitrogen dioxide and nitric acid, where most of the surface resistances are prescribed too. A detailed description of the submodel can be found in Kerkweg et al. (2006a, named DRYDEP). In COSMO/MESSy the dry deposition is applied (as described above) only as tracer tendency in the lowermost grid layer.

The necessary offline fields for the dry deposition parametrisation (e.g. soil pH, leaf area index, drag coefficient) are currently only available at a horizontal resolution of $0.5° \times 0.5°$ .

– JVAL

To calculate the photolysis rate coefficients the submodel JVAL is used, which is based on Landgraf and Crutzen (1998). The current version of this submodel is described by Sander et al. (2014). In COSMO/MESSy the required ozone input data, providing the ozone column above the model domain top, is downscaled from EMAC using the MMD (Multi Model Driver) submodels.

– LNOX

The submodel LNOX (described by Tost et al., 2007) calculates the $NO_x$ emissions due to lightning. However, up to now no detailed comparison of the results from the different lightning $NO_x$ parametrisations in COSMO/MESSy with observations has been conducted. This needs to be done in the near future. This is not relevant for this study, as, for comparison reasons, the downscaled lightning $NO_x$ fluxes (from EMAC) have been the means of choice.

– MECCA

The submodel MECCA (Module Efficiently Calculating the Chemistry of the Atmosphere, Sander et al., 2011) comprises the atmospheric reaction mechanism used to calculate the chemical kinetics. As described by Jöckel et al. (2015) the submodel was recently revised with updated rate coefficients according to the newest Jet Propulsion Laboratory (JPL) recommendations. For the simulations performed here the mechanism 'CCMI-base-01-tag.bat' is used. This mechanism includes the chemistry of ozone, methane and odd nitrogen. While alkynes and aromatics are not considered, alkenes and alkanes are considered up to $C_4$. We use the Mainz Isoprene Mechanism (MIM1, Pöschl et al., 2000) for the chemistry of isoprene and some non-methane hydrocarbons (NMHCs). The detailed mechanism is part of the Supplement.

– MSBM

For the consistent calculation of the heterogeneous reaction rates on polar stratospheric clouds (PSC) the Multiphase Stratospheric Box Model (MSBM, see Jöckel et al., 2010) is used. Additionally, this submodel determines the partitioning of $H_2O$ between gas- liquid- and ice-phase, which affects the hydrological cycle and feedbacks on the dynamics.

– OFFEMIS

For the emissions described by prescribed fluxes the submodel OFFEMIS is used (described as OFFLEM by Kerkweg




et al., 2006b). The prescribed fields are transformed on the computational grid using the submodel IMPORT (Kerkweg and Jöckel, 2015). Similar as in DDEP the emissions in COSMO/MESSy are applicable only as a tracer tendency.

– ONEMIS

The submodel ONEMIS (described as ONLEM by Kerkweg et al., 2006b) calculates different emission fluxes of selected chemical species on-line. In this study we use ONEMIS to calculate soil/biogenic emission of NO and biogenic emissions of isoprene ($C_5H_8$). For NO the algorithm is based on Yienger and Levy (1995) and on Guenther et al. (1995) for isoprene. The same data (for the leaf area index and the soil fertilizer classes) as for EMAC are used in COSMO/MESSy. These data have a resolution of $0.5°$ x $0.5°$ and should be updated to a higher resolution in the near future. For COSMO/MESSy only the option to add the emissions as tracer tendencies is available.

– SCAV

The scavenging of trace gases (and aerosols) by clouds and precipitation is treated by the submodel SCAV (Tost et al., 2006a, 2010). As COSMO/MESSy operates on shorter time steps, the equilibrium between gas and cloud phase can not be reached within each model time step in contrast to the EMAC application, where this can be considered a valid assumption. Therefore, additional tracers for the chemical species in the cloud phase (liquid and ice) have been added, which allow for transport of in-cloud tracers and consistent uptake (release) into (out of) the cloud droplets depending on the microphysical processes and thermodynamic conditions in the simulated clouds.

– TNUDGE

The submodel TNUDGE (Kerkweg et al., 2006b) allows a relaxation of tracers to specific mixing ratios and is mainly used for species with long but uncertain lifetimes, uncertain emission fluxes but well observed mixing ratios. In our simulations TNUDGE mainly prescribes $CH_4$, $CO_2$ and the CFCs mixing ratios at the surface. So far, the fields which are used in COSMO/MESSy by TNUDGE can be downscaled from EMAC using MMD submodels or imported using IMPORT.

*Acknowledgements.* M.Mertens acknowledges funding by the DLR project 'Verkehr in Europa'. C. Hofmann und A. Kerkweg additionally like to acknowledge funding by the German Ministry of Education and Research (BMBF) in the framework of the MiKlip (Mittelfristige Klimaprognose/Decadal Prediction) subproject FLAGSHIP (Feedback of a Limited-Area model to the Global-Scale implemented for HIndcasts and Projections, funding ID 01LP1127A). Furthermore, this work is based on work funded by the German Science Foundation (DFG) under the project name MACCHIATO (WE 2943/4-1). We thank M. Raschendorfer (DWD) for helpful comments on the turbulence parametrisation in COSMO. We also thank U. Blahak (DWD) and the COSMO-CLM community for their support. In addition we are thankful for very helpful comments from A. Lauer (DLR) which improved this manuscript.

We acknowledge the EBAS platform (ebas.nilu.no) for providing a broad range of observational data used in this study. We also acknowledge the World Ozone and Ultraviolet Radiation Data Centre (WOUDC) for the access to the vertical ozone profiles, retrieved from http://woudc.org. We further acknowledge the IAGOS-CARIBIC team for providing the aircraft in situ observations. Further we acknowledge the free use of tropospheric NO2 column data from the SCIAMACHY sensor from www.temis.nl.



Analysis and graphics of the used data was performed using the NCAR Command Language (Version 6.2.0) Software developed by UCAR/NCAR/CISL/TDD and available on-line: http://dx.doi.org/10.5065/D6WD3XH5. For the model development and the simulations presented here a lot of computational resources were needed. Therefore we acknowledge the computational resources provided by the Leibniz Supercomputing Centre (LRZ) in Garching and the German Climate Computing Centre (DKRZ) in Hamburg.



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




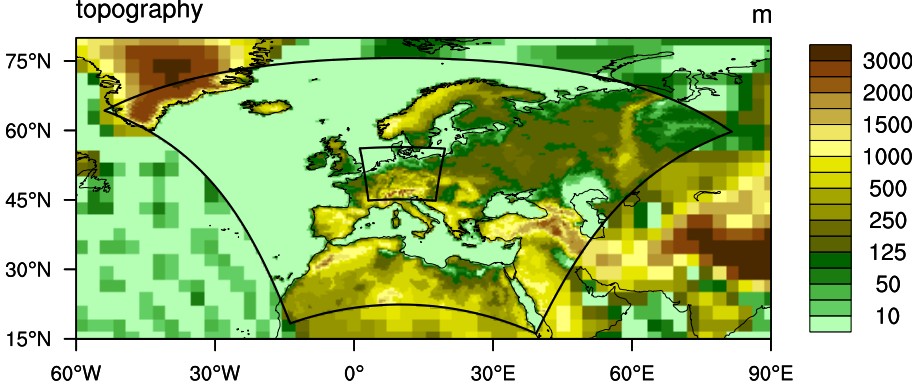

**Figure 1.** Computational domain of the COSMO(50km)/MESSy and COSMO(12km)/MESSy instances. Depicted is the topography (in m) in the resolution of the corresponding instance. Outside the COSMO(50km)/MESSy domain the values of EMAC are displayed. In both cases the whole computational domains, including the boundary zones, are shown.

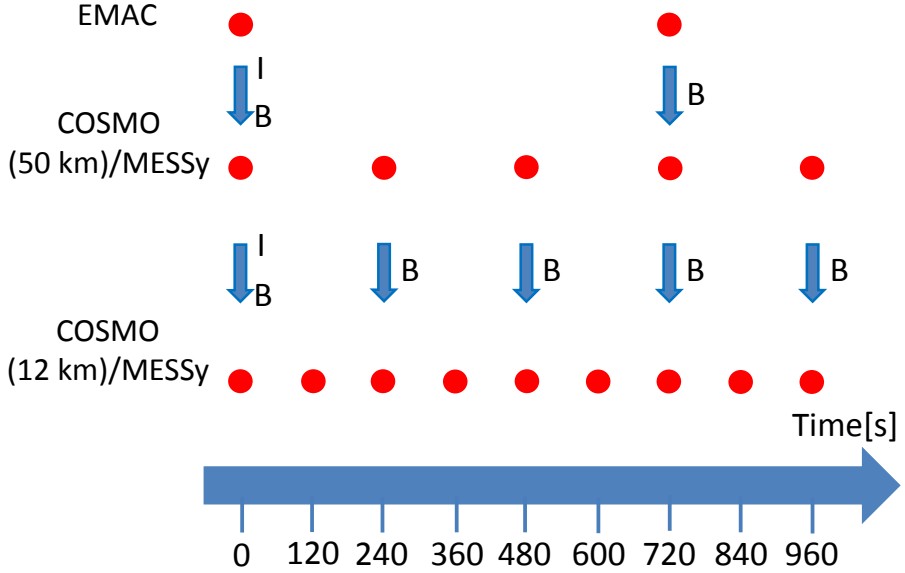

**Figure 2.** Illustration of the MECO(2) data exchange used in this study. The red circles indicate the time steps, the blue arrows indicate the data exchange. The exchange of initial data is marked with I, the exchange of boundary data with B.





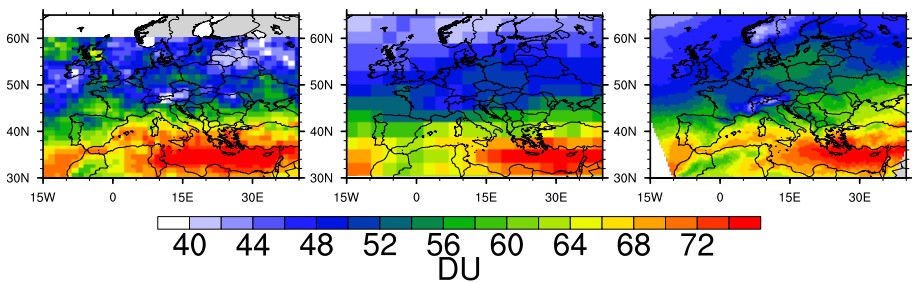

**Figure 3.** Tropospheric ozone columns in Dobson Units (DU) of OMI (left), EMAC (centre) and COSMO(50km)/MESSy (right) for June 2008. Please note, that the OMI-values are scaled with 1.45 for a better comparability (allowing the same colour bar).

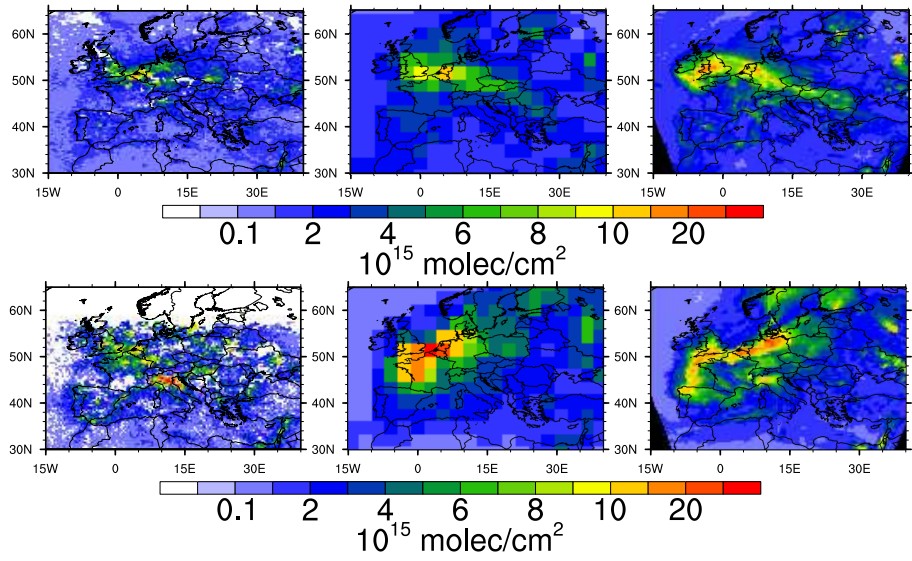

**Figure 4.** Tropospheric NO$_2$ columns (in $10^{15}$ molec cm$^{-2}$) of SCIAMACHY (left), EMAC (centre) and COSMO(50km)/MESSy (right) for (a) June and (b) December 2008.





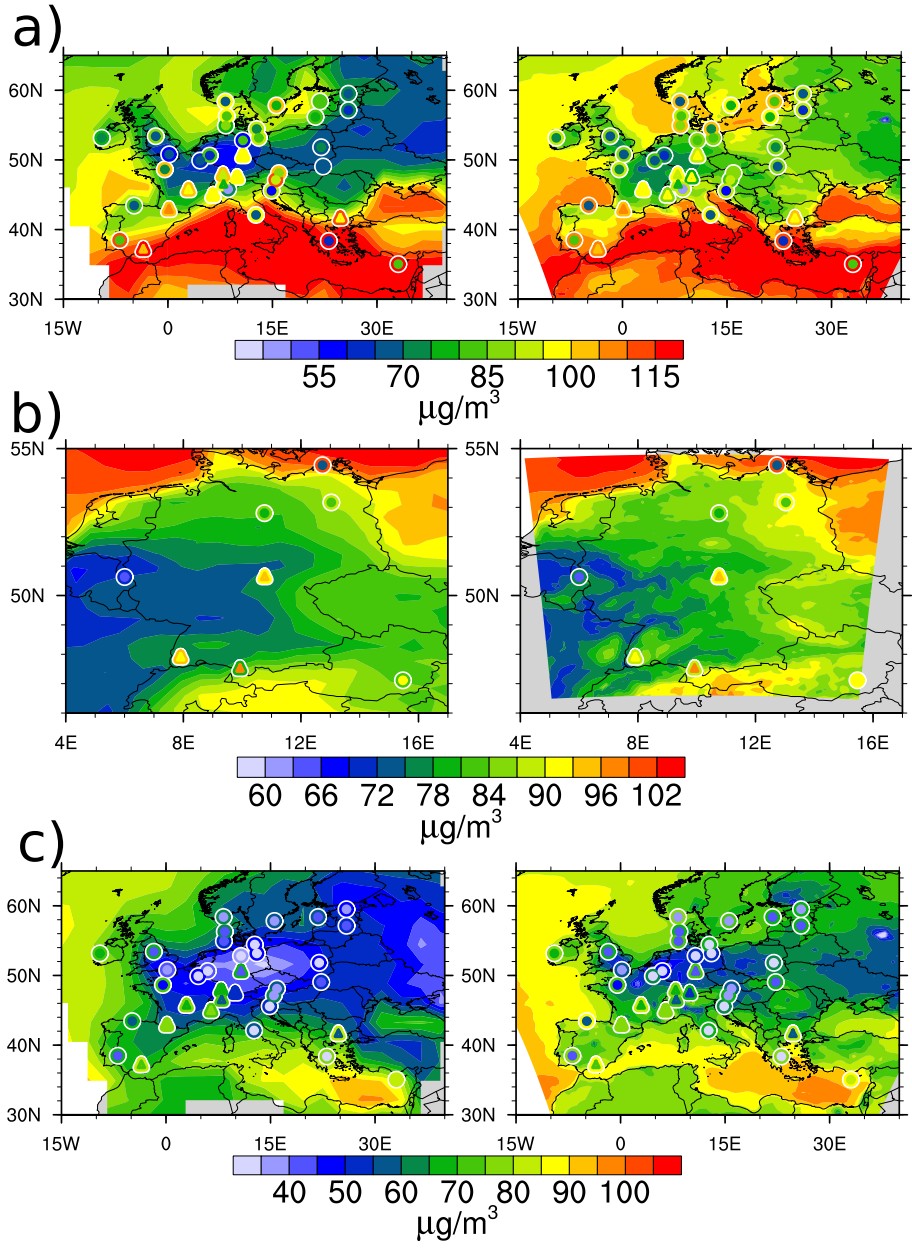

**Figure 5.** Monthly averaged ozone concentrations (µg m$^{-3}$) at the lowest model layer. The inner part of the coloured dots shows the monthly mean values measured at the corresponding stations, while the outer part depicts the simulated value corrected for the station elevation. Triangles indicate stations with an elevation higher than 800 m, circles stations below that height. (a) ozone concentration from EMAC (left) and COSMO(50km)/MESSy (right) in June 2008, (b) ozone concentration for COSMO(50km)/MESSy (left) and COSMO(12km)/MESSy (right) and (c) ozone concentration for EMAC (left) and COSMO(50km)/MESSy (right) in December 2008.



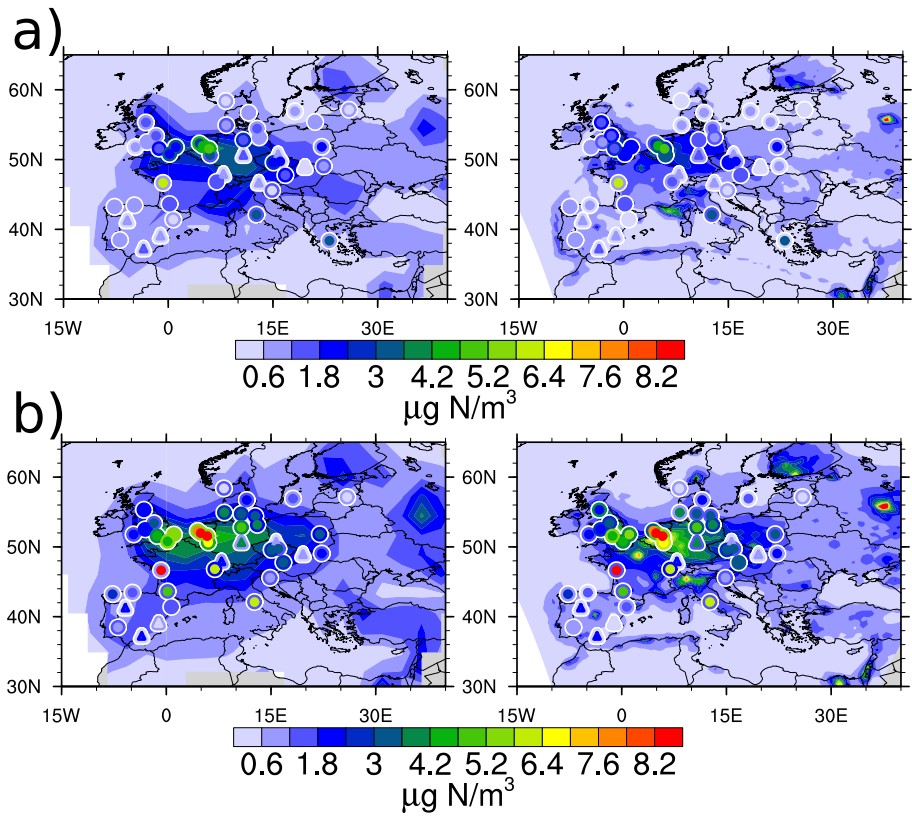

**Figure 6.** Monthly averaged nitrogen dioxide concentrations ($\mu g(N)\ m^{-3}$) at the lowest model layer in (a) June and (b) December 2008 from EMAC (left) and COSMO(50km)/MESSy (right). The inner part of the coloured dots shows the monthly mean values measured at the corresponding stations, while the outer part depicts the simulated value corrected for the station elevation. Triangles indicate stations higher than 800 m, circles stations below that height.

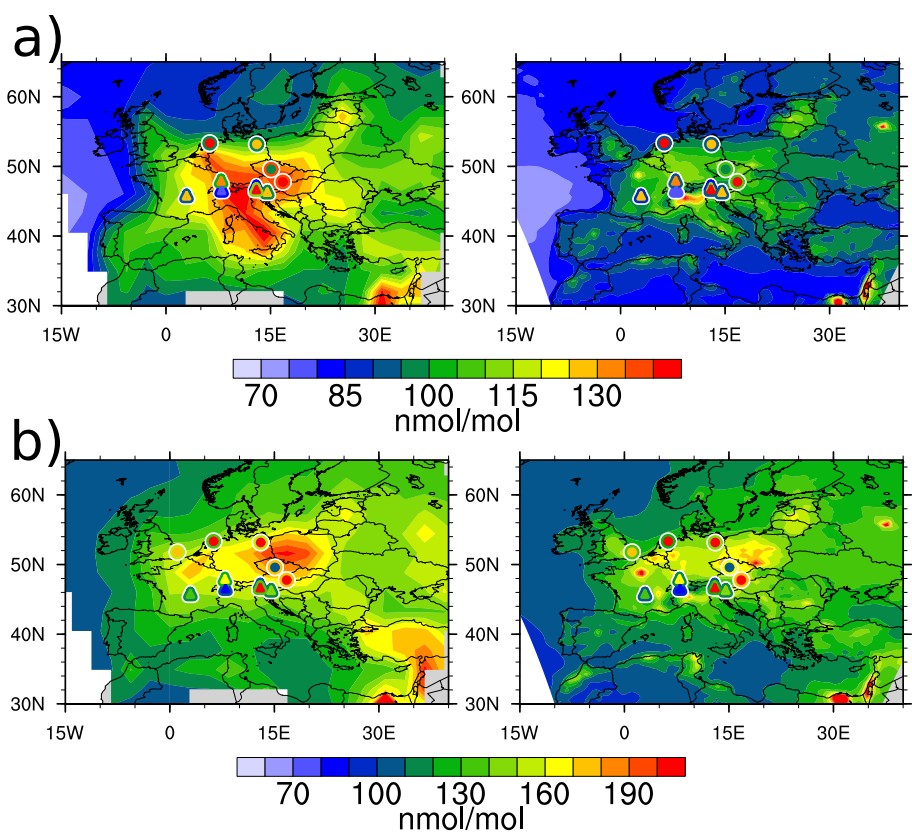

**Figure 7.** Monthly averaged carbon monoxide mixing ratios ($\mathrm{nmol\,mol^{-1}}$) at the lowest model layer in (a) June and (b) December 2008 from EMAC (left) and COSMO(50km)/MESSy (right). The inner part of the coloured dots shows the monthly mean values measured at the corresponding stations, while the outer part depicts the simulated value corrected for the station elevation. Triangles indicate stations higher than 800 m, circles stations below that height.





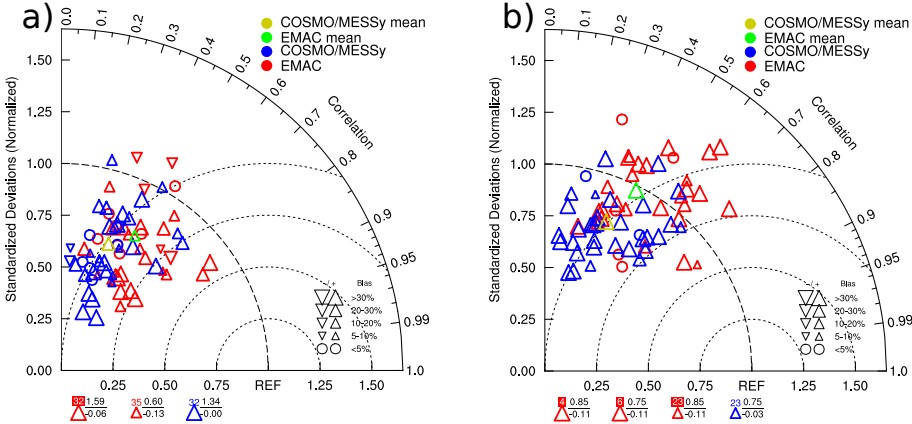

**Figure 8.** Taylor diagram of ground-level ozone concentrations for June (a) and December (b) 2008. The results for EMAC are shown in red, for COSMO(50km)/MESSy in blue. The mean over all stations is coloured in green for EMAC and in golden colour for COSMO/MESSy. The size of the symbols indicate the bias in percent; upward symbols signify a positive bias, downward symbols a negative bias. The symbols below the horizontal axis indicates the stations, which are out of range. The coloured number provides the number of the station, the upper black number depicts the normalized standardized deviation and the lower number shows the correlation coefficient at the station

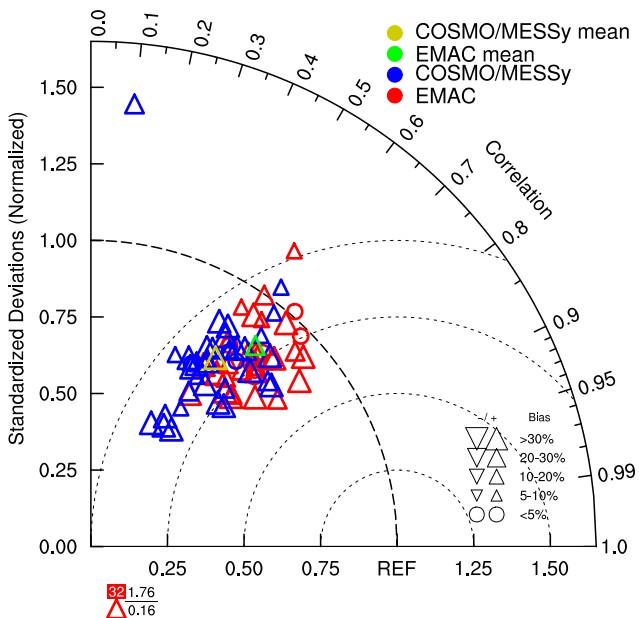

**Figure 9.** As Fig. 8 but for the whole year 2008.





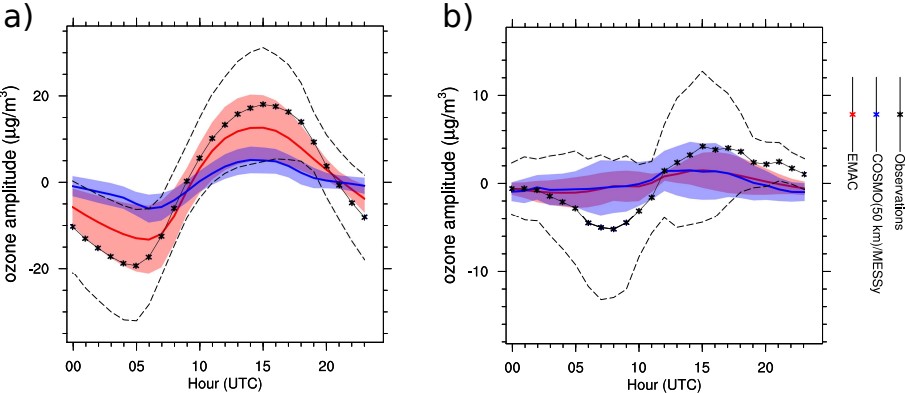

**Figure 10.** Diurnal cycle amplitude of ozone in µg m$^{-3}$ for (a) all non-mountain stations and (b) all mountain stations for June 2008. The observations are shown in black, while EMAC is shown in red and COSMO(50km)/MESSy in blue. The dashed lines indicate the standard deviation over all stations of the observations, while the coloured polygons display the standard deviation of the simulation data.

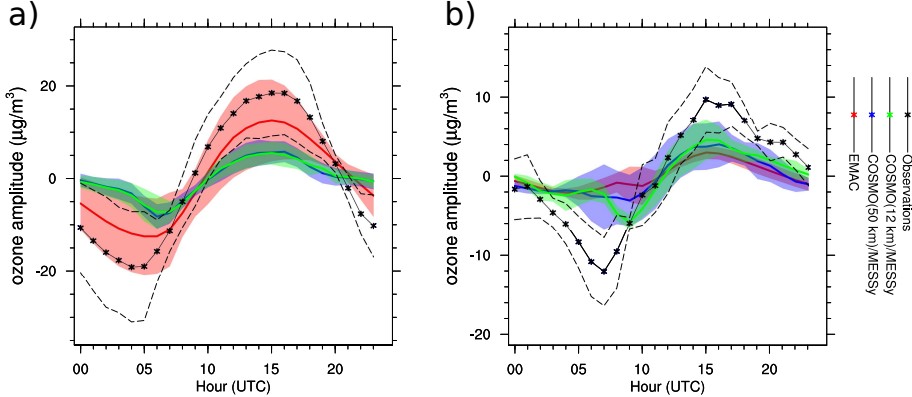

**Figure 11.** As Fig. 10 but for the subset of stations which are located in all three model instances.



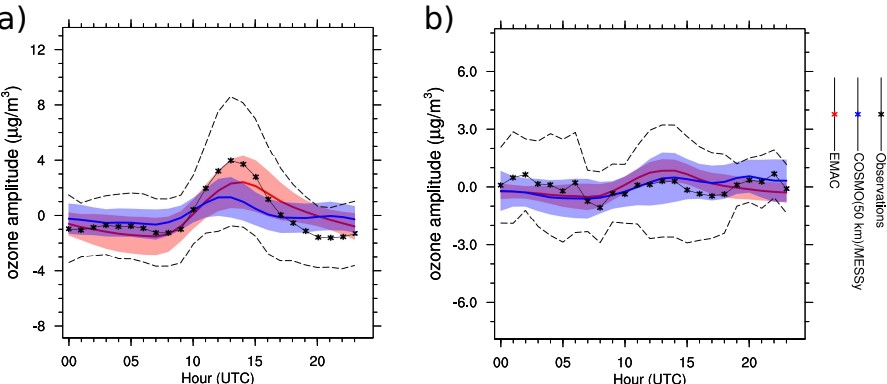

**Figure 12.** As Fig. 10 but for December 2008.

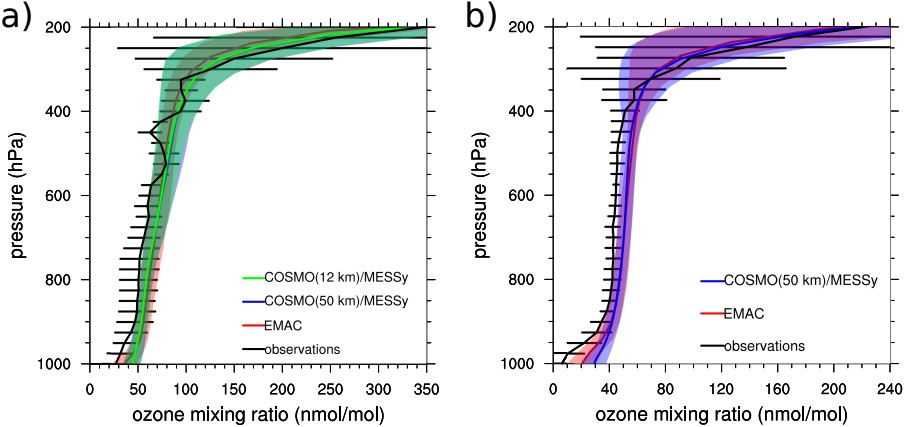

**Figure 13.** Vertical ozone profile (in $nmol\ mol^{-1}$) at Hohenpeissenberg (Germany) for (a) June, (b) December 2008. In (a) results for all EMAC and the two COSMO/MESSy instances are shown, while (b) shows the results for EMAC and COSMO(50km)/MESSy. The standard deviation of the temporal mean is indicated by the error bars for the observations and by the shaded area for the simulation data.





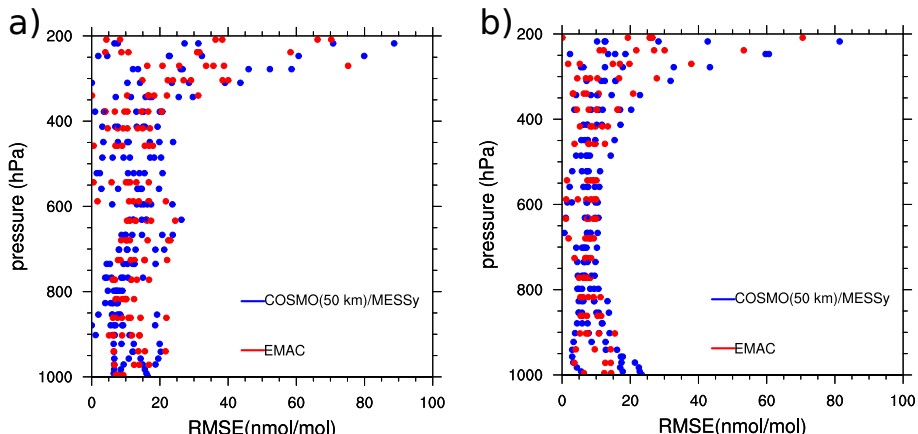

**Figure 14.** Vertical profile showing the RMSE of the model data compared to the ozone sonde data (in $\mathrm{nmol\ mol^{-1}}$) for (a) June 2008 and (b) December 2008.



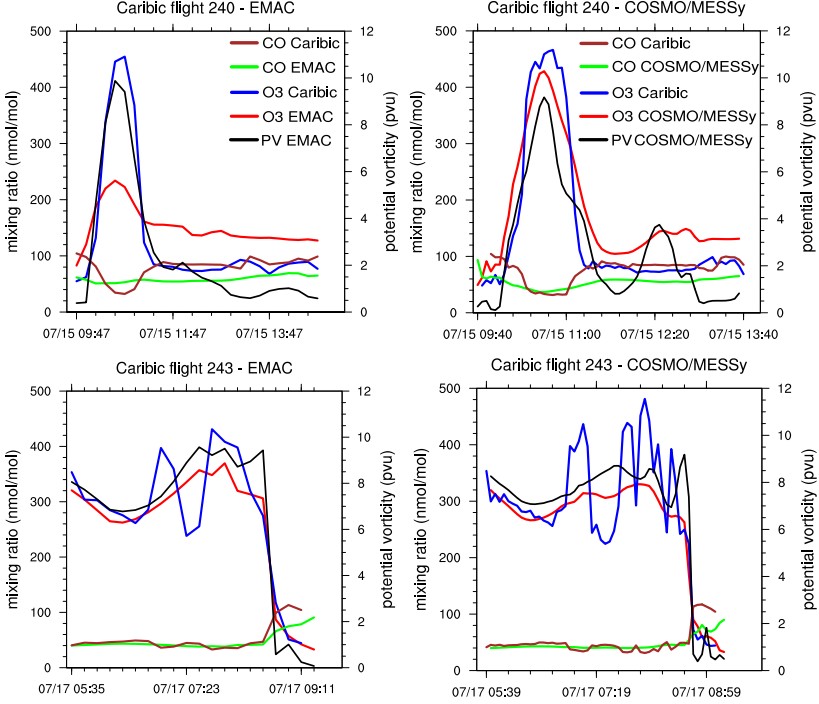

**Figure 15.** Comparison between IAGOS-CARIBIC measurements of ozone, carbon monoxide (left axis in $\mathrm{nmol\,mol^{-1}}$) for EMAC (left side) and COSMO(50km)/MESSy (right side). The upper row shows the results for the IAGOS-CARIBIC flight 240 and the lower row for the flight 243. For both models also the potential vorticity (right axis in PVU) is displayed as a proxy for tropospheric or stratospheric air masses.





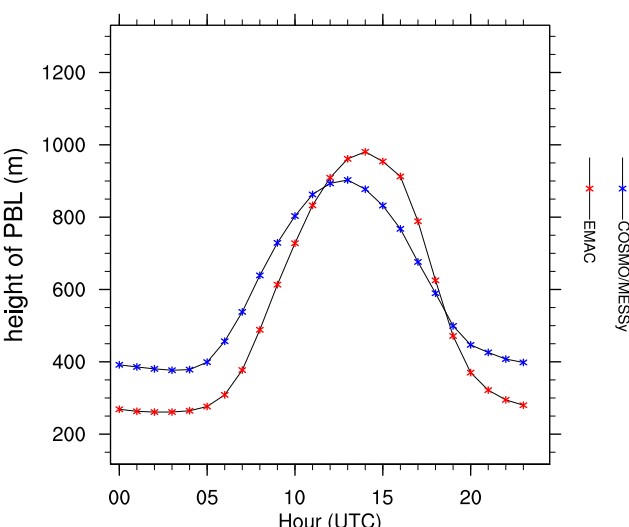

**Figure 16.** Height of the planetary boundary layer for June 2008 in m averaged over all non-mountain stations.





**Table 1.** Overview of the most important submodels applied in EMAC and COSMO/MESSy respectively. Both COSMO/MESSy instances use the same set of submodels. The complete list can be found in the Supplement (Sect. 6). MMD* comprise all MMD submodels.

| Submodel | EMAC | COSMO | short description | references |
|---|---|---|---|---|
| AEROPT | x | | calculation of aerosol optical properties | Dietmüller et al. (2016) |
| AIRSEA | x | x | exchange of tracers between air and sea | Pozzer et al. (2006) |
| CH4 | x | | methane oxidation and feedback to hydrological cycle | |
| CLOUD | x | | cloud parametrisation | Roeckner et al. (2006), Jöckel et al. (2006) |
| CLOUDOPT | x | | cloud optical properties | |
| CONVECT | x | | convection parametrisation | Tost et al. (2006b) |
| CVTRANS | x | x | convective tracer transport | Tost et al. (2010) |
| DRADON | x | x | emission and decay of $^{222}$Radon | Jöckel et al. (2010) |
| DDEP | x | x | dry deposition of aerosols and tracer | Kerkweg et al. (2006a) |
| E2COSMO | x | | additional ECHAM5 fields for COSMO coupling | Kerkweg and Jöckel (2012b) |
| GWAVE | x | | parametrisation of non-orographic gravity waves | Roeckner et al. (2003) |
| H2O | x | | stratospheric water vapour and its feedback | Jöckel et al. (2006) |
| JVAL | x | x | calculation of photolysis rates | Landgraf and Crutzen (1998), Jöckel et al. (2006) |
| LNOX | x | | $NO_x$-production by lighting | Tost et al. (2007), Jöckel et al. (2010) |
| MECCA | x | x | tropospheric and stratospheric gas-phase chemistry | Sander et al. (2011), Jöckel et al. (2010) |
| MMD* | x | x | coupling of EMAC and COSMO/MESSy (including libraries and all submodels) | Kerkweg and Jöckel (2012b) |
| MSBM | x | x | multiphase chemistry of the stratosphere | Jöckel et al. (2010) |
| OFFEMIS | x | x | prescribed emissions of trace gases and aerosols | Kerkweg et al. (2006b) |
| ONEMIS | x | x | on-line calculated emissions of trace gases and aerosols | Kerkweg et al. (2006b) |
| ORBIT | x | x | Earth orbit calculations | Dietmüller et al. (2016) |
| QBO | x | | Newtonian relaxation of the quasi-biennial oscillation (QBO) | Giorgetta and Bengtsson (1999), Jöckel et al. (2006) |
| RAD | x | | radiative transfer calculations calculation | Dietmüller et al. (2016) |
| S4D | x | x | diagnostic sampling along predefined tracks | Jöckel et al. (2010) |
| SCAV | x | x | wet deposition and scavenging of trace gases and aerosols | Tost et al. (2006a) |
| SCOUT | x | x | diagnostic sampling at predefined locations | Jöckel et al. (2010) |
| SEDI | x | x | sedimentation of aerosols | Kerkweg et al. (2006a) |
| SORBIT | x | x | sampling along sun synchronous satellite orbits | Jöckel et al. (2010) |
| SURFACE | x | | surface properties | Jöckel et al. (2015) |
| TAGGING | x | x | TAGGING of source attributions | Grewe et al. (2016) |
| TNUDGE | x | x | Newtonian relaxation of tracers | Kerkweg et al. (2006b) |
| TROPOP | x | x | diagnostic calculation of tropopause height and additional diagnostics | Jöckel et al. (2006) |





**Table 2.** Root-mean-square error (RMSE, in µg m$^{-3}$ for O$_3$ and NO$_2$, in nmol mol$^{-1}$ for CO) and normalized mean-bias error (MBE, in %) for EMAC and COSMO(50km)/MESSy (C(50km)/M) in comparison to ground-level observations. Shown are the values for June and December 2008. For the comparison always the height corrected values are used. The values are calculated from the monthly averaged values for all stations with observations for the given variable.

| | RMSE EMAC | RMSE C(50km)/M | MBE EMAC | MBE C(50km)/M |
|---|---|---|---|---|
| O$_3$ Jun. | 20.0 | 22.1 | 16.1 | 20.3 |
| O$_3$ Dec. | 19.4 | 27.5 | 34.7 | 54.7 |
| NO$_2$ Jun. | 1.18 | 1.13 | -17.6 | -33.8 |
| NO$_2$ Dec | 2.78 | 2.89 | -41.7 | -46.2 |
| CO Jun. | 42.8 | 47.3 | -20.2 | -28.0 |
| CO Dec | 57.7 | 63.8 | -20.1 | -24.8 |





**Table 3.** RMSE (in µg m$^{-3}$) and MBE (in %) for EMAC, COSMO(50km)/MESSy (C(50km)/M) and COSMO(12km)/MESSy (C(12km)/M) in comparison to ground-level observations. Shown are the values of ozone and nitrogen dioxide for June 2008. The values are calculated for the subset of measurement sites which are located in the COSMO(12km)/MESSy domain. The values are calculated from the monthly averaged values for all stations with observations for the given variable.

| | RMSE EMAC | RMSE C(50km)/M | RMSE C(12km)/M | MBE EMAC | MBE C(50km)/M | MBE C(12km)/M |
|---|---|---|---|---|---|---|
| $O_3$ Jun. | 14.9 | 12.3 | 11.4 | 10.1 | 3.94 | 6.54 |
| $NO_2$ Jun. | 0.805 | 0.865 | 0.846 | -10.6 | -28.8 | -29.0 |





**Table 4.** Mean values which are subtracted from the diurnal cycle (in μg m$^{-3}$) for EMAC and COSMO(50km)/MESSy (C(50km)/M). The given uncertainty is the standard deviation over all stations.

|  | June non-mountain | June mountain | December non-mountain | December mountain |
|---|---|---|---|---|
| EMAC | $88.8 \pm 19.2$ | $103.4 \pm 8.5$ | $57.5 \pm 11.6$ | $72.6 \pm 9.3$ |
| C(50km)/M | $95.3 \pm 12.1$ | $95.3 \pm 8.7$ | $68.2 \pm 9.7$ | $77.1 \pm 6.0$ |
| Observations | $74.2 \pm 11.4$ | $95.5 \pm 7.2$ | $39.1 \pm 13.2$ | $64.2 \pm 11.3$ |





**Table 5.** As Table 4 but only for the stations located in the COSMO(12km)/MESSy domain.

|  | June non-mountain | June mountain |
| --- | --- | --- |
| EMAC | $89.6 \pm 17.3$ | $99.1 \pm 1.6$ |
| C(50km)/M | $87.2 \pm 10.3$ | $88.3 \pm 8.2$ |
| C(12km)/M | $89.9 \pm 7.4$ | $89.4 \pm 0.9$ |
| Observations | $79.2 \pm 8.9$ | $94.4 \pm 2.2$ |





**Table 6.** Average values for June-August 2008 of the $CH_4$ mass ($M_{CH_4}$), the OH mass ($M_{OH}$) and the methane lifetime against OH ($\tau$) for EMAC, COSMO(50km)/MESSy (C(50km)/M) and COSMO(12km)/MESSy (C(12km)/M). All values are computed for the area of the COSMO(12km)/MESSy instance. The mass of $CH_4$ and OH are the time averaged values. The uncertainty range is the standard deviations with respect to time. The subscripts on the individual variables indicate the different vertical layers.

|  | EMAC | C(50km)/M | C(12km)/M |
| --- | --- | --- | --- |
| $M_{CH_{4850}}$ (Tg) | $0.973 \pm 0.011$ | $0.900 \pm 0.012$ | $0.916 \pm 0.012$ |
| $M_{OH_{850}}$ (kg) | $60.4 \pm 8.8$ | $46.9 \pm 5.3$ | $55.5 \pm 5.8$ |
| $\tau_{850}$ (a) | $2.73 \pm 0.46$ | $3.43 \pm 0.38$ | $2.90 \pm 0.29$ |
| $M_{CH_{4500}}$ (Tg) | $2.50 \pm 0.04$ | $2.45 \pm 0.04$ | $2.41 \pm 0.04$ |
| $M_{OH_{500}}$ (kg) | $192 \pm 15$ | $209 \pm 15$ | $212 \pm 15$ |
| $\tau_{500}$ (a) | $3.96 \pm 0.27$ | $3.54 \pm 0.22$ | $3.51 \pm 0.18$ |
| $M_{CH_{4200}}$ (Tg) | $2.12 \pm 0.04$ | $2.17 \pm 0.03$ | $2.11 \pm 0.03$ |
| $M_{OH_{200}}$ (kg) | $228 \pm 19$ | $248 \pm 24$ | $247 \pm 24$ |
| $\tau_{200}$ (a) | $12.4 \pm 1.0$ | $11.3 \pm 1.0$ | $11.2 \pm 1.1$ |





**Table 7.** As Table 6 but for the European area.

|  | EMAC | C(50km)/M |
| --- | --- | --- |
| $M_{\mathrm{CH_4}_{850}}$ (Tg) | $27.0 \pm 0.3$ | $26.7 \pm 0.3$ |
| $M_{\mathrm{OH}_{850}}$ (kg) | $1520 \pm 110$ | $1400 \pm 90$ |
| $\tau_{850}$ (a) | $2.71 \pm 0.16$ | $2.97 \pm 0.17$ |
| $M_{\mathrm{CH_4}_{500}}$ (Tg) | $69.1 \pm 1.0$ | $73.6 \pm 0.7$ |
| $M_{\mathrm{OH}_{500}}$ (kg) | $4620 \pm 400$ | $4630 \pm 400$ |
| $\tau_{500}$ (a) | $4.27 \pm 0.33$ | $4.50 \pm 0.36$ |
| $M_{\mathrm{CH_4}_{200}}$ (Tg) | $58.7 \pm 1.0$ | $58.7 \pm 0.7$ |
| $M_{\mathrm{OH}_{200}}$ (kg) | $5850 \pm 550$ | $5580 \pm 510$ |
| $\tau_{200}$ (a) | $12.8 \pm 1.3$ | $13.1 \pm 1.4$ |





**Table 8.** Area averaged ground-level concentrations (for a box from $5°W - 20°E$, $20°N - 55°N$) in $\mu g \, m^{-3}$ of various chemical species. The columns two to four display the values for COSMO(50km)/MESSy (C(50 km)/M), EMAC and COSMO(50km)/MESSy with changed temperature for the submodel MECCA C(50 km)/$M_{T*}$ respectively. The fifth column indicates, if the difference between EMAC and COSMO(50km)/MESSy are positive (+), negative (-), or if there is only a minor difference ($\approx$). The last column indicates the corresponding difference between COSMO(50km)/MESSy and COSMO(50km)/MESSy with changed temperature field for MECCA.

|  | C(50 km)/M | EMAC | C(50 km)/$M_{T*}$ | diff 2 and 1 | diff 3 and 1 |
|---|---|---|---|---|---|
| $HO_2$ | 0.00453 | 0.00563 | 0.00492 | + | + |
| OH | $4.53 \cdot 10^{-5}$ | $5.67 \cdot 10^{-5}$ | $4.75 \cdot 10^{-5}$ | + | + |
| $CHBr_3$ | 0.00388 | 0.00402 | 0.00387 | + | $\approx$ |
| $CH_3Br$ | 0.0308 | 0.0320 | 0.0308 | + | $\approx$ |
| $CH_3I$ | 0.00261 | 0.00412 | 0.00261 | + | $\approx$ |
| $NO_3$ | 0.00866 | 0.00914 | 0.0126 | + | + |
| $NH_3$ | 2.04 | 4.00 | 2.03 | + | $\approx$ |
| NO | 0.178 | 0.401 | 0.164 | + | - |
| $NO_2$ | 3.51 | 5.25 | 3.55 | + | $\approx$ |
| $C_5H_8$ | 0.0855 | 0.144 | 0.0818 | + | - |
| HCHO | 0.85 | 1.14 | 0.938 | + | + |
| CO | 149 | 169 | 149 | + | $\approx$ |
| $O_3$ | 111 | 99.0 | 114 | - | + |