# Peer review of "The 1-way on-line coupled model system MECO(n) – Part 4: Chemical evaluation (based on MESSy v2.52)"

_Geoscientific Model Development, 2015_

## Referee Comment (RC1) · Anonymous Referee #1 · 24 May 2016

General:

The paper is the fourth part of a paper series about MECO(n), an on-line coupled model version of COSMO-CLM and EMAC. In this paper tropospheric chemistry is discussed in detail for the first time. The model results are compared to different observations. Potential problems in the model system are discussed in a sufficient way. The paper is well written and I only suggest some minor corrections. Therefore the paper should be published in GMD.

Minor corrections:

As this is the fourth part of a paper series about MECO(n), can you please add a few

words about part 1 to 3 in the introduction?

p.3, line 30: Please tell the reader where this emission data set is published or described. Published elsewhere is not enough.

Chapter 2.1: In the introduction you write that one of the advantages of MECO(n) is that for standard CCMs "current computational resources pose an upper limit". Here you write that you have to exchange data between the different instances (which also costs time) and that there are additional waiting times for data exchange. Can you give an estimation how much time you save in total compared to doing a high resolution EMAC simulation?

p.6, line 24: You say that you don't consider Averaging Kernels (AK) in your comparisons and therefore you focus on horizontal patterns. This is only possible if AKs do not change in horizontal direction. Probably that is not a problem but can you please check?

p.7, line 22: Here it may sound as the cold bias is due to the coupling but I guess it is the same known problem in COSMO-CLM you mention on page 14. Please also add a short remark here.

p.8, section 4.2: Please clarify where height corrected values are used and where not. Especially in the beginning of the section I don't know if height corrected values are used or not.

In figure 6 the highest values for the height corrected values seem to appear in Belgium/Netherlands and near Nantes (France). Both areas seem to be rather flat. Can you explain why you have the highest corrections there?

Height correction: Can you please give a short description (maybe in the appendix) how the height corrected values are calculated? Is it just scaling with pressure or is it something more complicated?

p. 19: MECCA: Can you please specify which version of the recommendations from

JPL you used? Meanwhile the newest version is from 2015, which (according to your supplement) is not used.

Technical/Typos:

p.1, line 7: "... and a one ..."; change to "... and one ..."

p.3, line 25; p.8, line 33: line to long (happens several times)

p.8, line 5: "we are do not present" → we do not present

p.8, line 15: Atlantic sea → Atlantic ocean

p.9, line 28: located in?

p.13, line 13: mixing ratios

---

## Referee Comment (RC2) · Anonymous Referee #2 · 13 Jul 2016

General comments:

This paper describes a new nested, coupled system for chemistry climate modelling, which has online coupling of the nested grids. This paper evaluates the gas phase tropospheric chemistry in this new model in comparison to the coarser resolution model and to observations. I think this paper is successful in its aims of describing and evaluating the new model. Subject to my minor comments below, I recommend this paper is accepted.

Specific comments:

A point with respect to rainbow colour scales – there is a good argument for not using

them, as they artificially distort the field that they are visualizing, and they also cause problems for people with colour blindness. I recommend changing the rainbow colour scales to ones that do not suffer from these flaws. See here for more about this: http://www.climate-lab-book.ac.uk/2014/end-of-the-rainbow/

P1 line 21: This sentence is awkward and difficult to read: Especially, as some of the relevant processes (for example tropospheric ozone chemistry) are non-linear, it is desirable to resolve smaller scales, since with finer resolution the capabilities of chemistry-climate models in simulating species like ozone or nitrogen dioxide can be enhanced. Possible alternative: It is desirable to resolve smaller scales because finer resolution chemistry-climate models can simulate species like ozone or nitrogen dioxide better, as some of the relevant processes are non-linear (for example tropospheric ozone chemistry).

P7L4: This sentence is awkward and hard to understand: To allow for a fair comparison between EMAC and COSMO/MESSy always the value of the model layer which is nearest to the elevation of the station is selected. Suggested alternative: To allow for a fair comparison between EMAC and COSMO/MESSy, the value of the model layer which is nearest to the elevation of the station is always selected.

P7L32: "The maximum over the Mediterranean sea is underestimated in COSMO(50km)/MESSy. EMAC simulates a higher extend of this maximum, which better corresponds to the satellite measurements." The word "extend" should be "extent". I also don't agree with this when I look at fig 3. What is meant by a "higher extent"? Higher in altitude – which we cannot see in this figure? Higher latitudes – which doesn't agree with what I can see from the figure? Larger figures with a better colour scale may help to show the reader what you are trying to convey here. The same applies to the following statement about the Alps, as it's hard to see details over such a small region.

Sec 4.1: How good are the satellite retrievals? Is there any bias that may account for some of the differences – eg a difference between the land and sea? I am not an

expert in satellite retrievals, so I think a sentence or two about whether there are any biases in the satellite data would be helpful here.

Sec 4.2.1 on Taylor diagrams. I don't think you explain anywhere what the x axis is on the Taylor diagram. Have I also understood correctly that the EMAC model does better according to the metrics described in this section than the COSMO/MESSy model? If this is the case then maybe some commentary to explain why this is would be good here. Or to refer forward to a section where you discuss this.

Sec 4.5: I assume the unit "a" means "annum". It took me a few seconds to work this out, and it isn't explicitly stated anywhere. I would have uses "years" or abbreviated to "y" or "yr", as this seems to be the convention in the literature. I don't know if GMD have a policy on this. Unless the reader has a good feel for what numbers to expect for the methane lifetime, the numbers in this section are not very helpful, in my opinion. Most non-specialists will simply know that the methane lifetime is approx 10 years globally, however the numbers in this section are very different to that. Some context or literature values would help here.

Technical corrections:

Abstract, Line 15: Change to: "In comparison with observations, both EMAC and COSMO/MESSy show strengths and weaknesses."

P2L18 – "consistence" should be "consistency "

P4L15 remove second comma: "Thus, it is desirable that all. . ."

P7L5: Rearrange to: This is very important, especially in mountainous terrain, as COSMO/MESSy resolves the topography much better. B7L8: suggest adding a comma before the word measurements to make this sentence a bit clearer

Caption fig 5: please specify that the middle row is for June 2008

P9L28: missing "in"

P12L10: Remove first comma: "In order to check if the vertical distribution of ozone is well simulated, . . ."

P16L8: Remove first comma: "To investigate if we can. . ." P16L16: "In addition, also" – you don't need both of these terms in this sentence P16L17: "of" should be "off" P16L28: remove second comma: "Furthermore, it is well known that the. . ."

P17L15: The word "especially" seems a bit out of place to me here. Suggest "Particularly" instead. P17L23: Again, "especially" seems out of place here. Suggest "particularly," or "in particular,". Later in this sentence "as by the coarser" should be "than by the coarser"

P17L26: "good" should be "well" P17L27: Another sentence beginning with especially – perhaps you wish to keep this one, however I'd remove the word as the sentence works as well without it. P17L28: Sentence starting with "The comparison" – remove the first comma. The final "to" should be "too". P17L32: remove first comma

---

## Author Comment (AC1) · 9 Aug 2016

We thank referee#1 for the very helpful comments which helped to improve the manuscript. Here are our replies:

- *The paper is the fourth part of a paper series about MECO(n), an on-line coupled model version of COSMO-CLM and EMAC. In this paper tropospheric chemistry is discussed in detail for the first time. The model results are compared to different observations. Potential problems in the model system are discussed in a sufficient way. The paper is well written and I only suggest some minor corrections. Therefore the paper should be published in GMD*

  Reply: We thank referee#1 for these very positive and encouraging comments.

- *As this is the fourth part of a paper series about MECO(n), can you please add a few words about part 1 to 3 in the introduction?*

  Reply: This is indeed a very good point. The corresponding publications of part 1–3 were mentioned in the introduction, but they are not highlighted in detail. We rephrased the introduction slightly to highlight the contributions of part 1–3 in more detail.

- *p.3, line 30: Please tell the reader where this emission data set is published or described. Published elsewhere is not enough.*

  Reply: The mentioned study is not finished yet, therefore we can't give a reference here. This dataset, however, is not used in the present study. Here the 'MACCIty' emission scenario (Granier et al., 2011) is used. We rephrased the sentence slightly to make this more clear.

- *In the introduction you write that one of the advantages of MECO(n) is that for standard CCMs "current computational resources pose an upper limit". Here you write that you have to exchange data between the different instances (which also costs time) and that there are additional waiting times for data exchange. Can you give an estimation how much time you save in total compared to doing a high resolution EMAC simulation?*

  Reply: That is indeed a good question. Actually the computational time (and especially the time needed for exchange of the date) heavily depends on the network of the computing system. In addition also the amount of core per node (which defines the possibilities to distribute the different tasks on the nodes) influence the computational resources needed for MECO(n). For this reasons we hesitated to discuss this in more detail in the manuscript.

  As a rough estimate EMAC at T42 resolution (with 31 vertical layers) needs around 130 node-h per year on 'mistral' at the Deutsches Klimarechenzentrum. The resolution of COSMO is around 6 times higher, as the resolution of EMAC, which ends up in a multiplication of the computation time with a factor of 36 (assuming perfect scaling with the increased

amount of gridboxes). In addition the time-step must be decreased by a factor of 3–4 (for further calculations we assume 3). This gives a computational demand of EMAC at COSMO resolution of roughly 14000 node-h per year ($130 \cdot 36 \cdot 3$). The nested set-up as described in this study with one instance over Europe needs roughly 3200 node-h per year on Mistral. Please keep further in mind, that COSMO features a finer vertical resolution (40 instead of 31 vertical levels) and that one timestep of COSMO is 'more expensive' than one timestep of EMAC, as for example a much more detailed land-model is used by COSMO in comparison to EMAC. Thus there is a benefit of a factor of four based on this estimates.

- *p.6, line 24: You say that you don't consider Averaging Kernels (AK) in your comparisons and therefore you focus on horizontal patterns. This is only possible if AKs do not change in horizontal direction. Probably that is not a problem but can you please check?*

Reply: Referee#1 is totally right with this remark. For SCIAMACHY Blond et al. (2007) compared model results with and without averaging kernel with satellite measurements and did not find huge differences (compare Fig. 5 b and d in Blond et al., 2007). For OMI differences between the diagnosed tropopause are likely more problematic than not considering AKs (see also discussion by Righi et al., 2015; Jöckel et al., 2016). This differences can also change horizontal patterns, which we overlooked. We therefore decided to rephrase the sentence:

**Therefore, only a qualitative comparison of the data is possible. A quantification of biases is rather based on the comparison with the ground-level observations.**

- *p.7, line 22: Here it may sound as the cold bias is due to the coupling but I guess it is the same known problem in COSMO-CLM you mention on page 14. Please also add a short remark here.*

Reply: We thank referee#1 for this remark. A short note, similar as on page 14, is added in the revised manuscript.

- *p.8, section 4.2: Please clarify where height corrected values are used and where not. Especially in the beginning of the section I don't know if height corrected values are used or not.*

Reply: We added a note that all metrics are based on the height corrected values.

- *In figure 6 the highest values for the height corrected values seem to appear in Belgium/Netherlands and near Nantes (France). Both areas seem to be rather flat. Can you explain why you have the highest corrections there?*

Reply: We do not see a large difference between the 'height corrected' and the uncorrected values at these stations. Nevertheless we think that referee#1 might refer to the large differences between the values displayed by the inner and the outer circle. These differences show that much higher

values are measured (inner circle) than simulated by the model (outer circle, 'height corrected'). Likely these differences correspond to large local sources which are not well represented in the used emission database. We improved the paragraph about height correction (see below).

- *Height correction: Can you please give a short description (maybe in the appendix) how the height corrected values are calculated? Is it just scaling with pressure or is it something more complicated?*

  Reply: We rephrased the description of Section 3 regarding the height correction to explain the procedure. As no inter- or extrapolation of the model results is performed we think that this description is sufficient:

  **To allow for a fair comparison between EMAC, COSMO/MESSy and the observations a 'height correction' of the model results from EMAC and COSMO/MESSy is applied. For the EMAC data the geometric height of each station is compared with the geopotential height of the individual model levels at the corresponding gridbox in which the station is located. For the COSMO data the procedure is analogue to EMAC, but the height of the model level instead of the geopotential height is chosen. We pick the model results at the vertical level, where the geopotential height (EMAC)/model level height (COSMO) is nearest to the geometric height of the station. No interpolation of the model results between different levels is performed. However, this option only works, if the station is located higher than the ground of the lowest model layer. In the opposite case, the values of the lowest model layer are chosen and no extrapolation of the simulated data is performed. This height correction is very important, especially over mountainous terrain, as the topography is much finer resolved by COSMO/MESSy. In other words, if the observations would always be compared to the model values at the lowest model level, COSMO/MESSy would outperform EMAC solely because of the finer resolved topography. The usage of these height corrected values is indicated in the corresponding sections.**

- *p. 19: MECCA: Can you please specify which version of the recommendations from JPL you used? Meanwhile the newest version is from 2015, which (according to your supplement) is not used.*

  Reply: We used the version of the evaluation cycle 17 (Sander et al., 2011), not the newest from cycle 18. A corresponding note is added in the revised manuscript.

- *p.1, line 7: "... and a one ..."; change to "... and one ..."*

  Reply: Done.

- *p.3, line 25; p.8, line 33: line to long (happens several times)*

Reply: We changed the long terms COSMO(12km)/MESSy and COSMO(50km)/MESSy to shorter abbreviations CM12 and CM50 in order to overcome some of the problems.

- *p.8, line 5: "we are do not present" → we do not present*

  Reply: Done.

- *p.8, line 15: Atlantic sea → Atlantic ocean*

  Reply: Done.

- *p.9, line 28: located in?*

  Reply: Yes indeed. Fixed.

- *p.13, line 13: mixing ratios*

  Reply: Done.

**References**

Blond, N., Boersma, K. F., Eskes, H. J., van der A, R. J., Van Roozendael, M., De Smedt, I., Bergametti, G., and Vautard, R.: Intercomparison of SCIAMACHY nitrogen dioxide observations, in situ measurements and air quality modeling results over Western Europe, Journal of Geophysical Research: Atmospheres, 112, n/a–n/a, 10.1029/2006JD007277, http://dx.doi.org/10.1029/2006JD007277, d10311, 2007.

Granier, C., Bessagnet, B., Bond, T., D'Angiola, A., van der Gon, H. D., Frost, G., Heil, A., Kaiser, J., Kinne, S., Klimont, Z., Kloster, S., Lamarque, J.-F., Liousse, C., Masui, T., Meleux, F., Mieville, A., Ohara, T., Raut, J.-C., Riahi, K., Schultz, M., Smith, S., Thompson, A., Aardenne, J., Werf, G., and Vuuren, D.: Evolution of anthropogenic and biomass burning emissions of air pollutants at global and regional scales during the 1980 - 2010 period, Climatic Change, 109, 163–190, 2011.

Jöckel, P., Tost, H., Pozzer, A., Kunze, M., Kirner, O., Brenninkmeijer, C. A. M., Brinkop, S., Cai, D. S., Dyroff, C., Eckstein, J., Frank, F., Garny, H., Gottschaldt, K.-D., Graf, P., Grewe, V., Kerkweg, A., Kern, B., Matthes, S., Mertens, M., Meul, S., Neumaier, M., Nützel, M., Oberländer-Hayn, S., Ruhnke, R., Runde, T., Sander, R., Scharffe, D., and Zahn, A.: Earth System Chemistry integrated Modelling (ESCiMo) with the Modular Earth Submodel System (MESSy) version 2.51, Geoscientific Model Development, 9, 1153–1200, 10.5194/gmd-9-1153-2016, http://www.geosci-model-dev.net/9/1153/2016/, 2016.

Sander, S. P., Abbatt, J., Barker, J. R., Burkholder, J. B., Friedl, R. R., Golden, D. M., Huie, R. E., Kolb, C. E., Kurylo, M. J., Moortgat, G. K., Orkin, V. L., and Wine, P. H.: Chemical Kinetics and Photochemical Data for Use in Atmospheric Studies, JPL Publication 10-6, Jet Propulsion Laboratory, Pasadena, available at: http://jpldataeval.jpl.nasa.gov (last access: 15. July 2016), 2011.

Righi, M., Eyring, V., Gottschaldt, K.-D., Klinger, C., Frank, F., Jöckel, P., and Cionni, I.: Quantitative evaluation of ozone and selected climate parameters in a set of EMAC simulations, Geosci. Model Dev., 8, 733–768, 10.5194/gmd-8-733-2015, http://www.geosci-model-dev.net/8/733/2015/, 2015.

---

## Author Comment (AC2) · 9 Aug 2016

We thank referee#2 for the very helpful comments which helped to improve the manuscript. Here are our replies:

- *This paper describes a new nested, coupled system for chemistry climate modelling, which has online coupling of the nested grids. This paper evaluates the gas phase tropospheric chemistry in this new model in comparison to the coarser resolution model and to observations. I think this paper is successful in its aims of describing and evaluating the new model. Subject to my minor comments below, I recommend this paper is accepted*

  Reply: We thank referee#2 for these very positive and encouraging comments.

- *A point with respect to rainbow colour scales – there is a good argument for not using them, as they artificially distort the field that they are visualizing, and they also cause problems for people with colour blindness. I recommend changing the rainbow colour scales to ones that do not suffer from these flaws. See here for more about this: http://www.climate-lab-book.ac.uk/2014/end-of-the-rainbow/*

  Reply: We agree with referee#2 that the rainbow colour scale has some problems. For the revised manuscript we changed the colour scale of the figures which used the rainbow scale before.

- *P1 line 21: This sentence is awkward and difficult to read: Especially, as some of the relevant processes (for example tropospheric ozone chemistry) are non-linear, it is desirable to resolve smaller scales, since with finer resolution the capabilities of chemistry-climate models in simulating species like ozone or nitrogen dioxide can be enhanced. Possible alternative: It is desirable to resolve smaller scales because finer resolution chemistry-climate models can simulate species like ozone or nitrogen dioxide better, as some of the relevant processes are non-linear (for example tropospheric ozone chemistry).*

  Reply: We thank referee#2 for this comment. We adopted the suggested change.

- *P7L4: This sentence is awkward and hard to understand: To allow for a fair comparison between EMAC and COSMO/MESSy always the value of the model layer which is nearest to the elevation of the station is selected. Suggested alternative: To allow for a fair comparison between EMAC and COSMO/MESSy, the value of the model layer which is nearest to the elevation of the station is always selected.*

  Reply: We thank referee#2 for this comment. According the comments from referee#1 the whole paragraph was rephrased in order to explain the 'height correction' in more detail.

- *P7L32: "The maximum over the Mediterranean sea is underestimated in COSMO(50km)/MESSy. EMAC simulates a higher extend of this maximum, which better corresponds to the satellite measurements." The word*

*"extend" should be "extent". I also don't agree with this when I look at fig 3. What is meant by a "higher extent"? Higher in altitude – which we cannot see in this figure? Higher latitudes – which doesn't agree with what I can see from the figure? Larger figures with a better colour scale may help to show the reader what you are trying to convey here. The same applies to the following statement about the Alps, as it's hard to see details over such a small region.*

Reply: Referee#2 is right with the fact, that our sentence was unclear. We rephrased this part for the revised manuscript:

**The overall patterns of all three ozone columns look very similar with a strong north-south gradient. Investigating into more detail, some differences are apparent. COSMO simulates the maximum ozone column mainly along the coastline of Turkey. Compared to this the maximum in EMAC extents further to the West and South. This corresponds better to the satellite measurements, which show the largest values in the whole southeastern part of the Mediterranean Sea.**

Further the colour scale was changed, as mentioned above and the panelling of the figure was changed.

- *Sec 4.1: How good are the satellite retrievals? Is there any bias that may account for some of the differences – a difference between the land and sea? I am not an expert in satellite retrievals, so I think a sentence or two about whether there are any biases in the satellite data would be helpful here.*

Reply: We add a note about the problems with the satellite retrievals in Sec. 3 where the data are described. We felt that the discussion of these biases is more suitable in this section than in Sect. 4.1. A reference to this discussion was added in Sect. 4.1.

- *Sec 4.2.1 on Taylor diagrams. I don't think you explain anywhere what the x axis is on the Taylor diagram.*

Reply: We rephrased the first part of the description to better explain the meaning of the x-axis.

**For a more quantitative comparison, Taylor diagrams (details are given by Taylor, 2001) are calculated. These diagrams combine the (normalised) standard deviation (as radius) and the correlation between the observed and the simulated time series (as angle). The observational reference point is marked with REF on the x-axis. The calculations are based on hourly averaged model output and observations, respectively. The bias in percent between the simulated and observed ozone concentration is displayed by the size of the symbols. The dashed circles indicate the root mean square error. Again, only the height corrected values are used, which improve the results of EMAC**

**considerably. The Taylor diagrams for the uncorrected cases are part of the Supplement (Sect. S1.4).**

- *Have I also understood correctly that the EMAC model does better according to the metrics described in this section than the COSMO/MESSy model? If this is the case then maybe some commentary to explain why this is would be good here. Or to refer forward to a section where you discuss this*

Reply: Yes indeed, EMAC performs better according to the described metrics. This is likely due to the problems with the diurnal cycle in COSMO. In the revised manuscript we added a short note with references to the next sections.

**The overall better results for EMAC compared to COSMO are likely caused by the deficits in the representation of the diurnal cycle in COSMO as discussed in Sect. 4.2.2. A more detailed discussion about potential reasons for this is provided in Sect. 5.**

- *Sec 4.5: I assume the unit "a" means "annum". It took me a few seconds to work this out, and it isn't explicitly stated anywhere. I would have uses "years" or abbreviated to "y" or "yr", as this seems to be the convention in the literature. I don't know if GMD have a policy on this.*

Reply: Yes, 'a' stands for the latin 'annum'. We know that in many other publications the abbreviation 'y' or 'yr' is used. In the informative annex C, the International Standard ISO 80000-3 proposes the symbol 'a ' to represent a year of either 365 or 366 days. Also the IUPC recommend the usage of 'a' (http://media.iupac.org/publications/books/gbook/IUPAC-GB3-2ndPrinting-Online-22apr2011.pdf). As GMD points to this document in the guidelines we think that 'a' is the right choice.

- *Unless the reader has a good feel for what numbers to expect for the methane lifetime, the numbers in this section are not very helpful, in my opinion. Most non-specialists will simply know that the methane lifetime is approx 10 years globally, however the numbers in this section are very different to that. Some context or literature values would help here.*

Reply: We agree with referee#2 on this point. The values we are showing are only for a part of the globe and are not comparable to 'typical' numbers in the literature. We use the lifetime simply as a proxy for the tropospheric oxidation capacity to check if this capacity changes between both models. In the revised manuscript we added a more detailed note on this.

**As shown by Jöckel et al. (2016) the methane lifetime against OH of the *RC1SD-base-10a* simulation, which has a very similar set-up as used in the present study (see Sect. 2.3), is around 7.7 a for the year 2008. As analysed in detail by Jöckel et al. (2016) this is at the lower end compared to results from other models which are mainly in the range from 8–9 a. The values we present**

here are not directly comparable to these global estimates of the methane lifetime, as we calculate the lifetime only for a part of the globe. Here, for a more detailed comparison of the results from EMAC and the two COSMO/MESSy instances we further calculate the lifetime separately for three different vertical layers of the atmosphere: From the ground to 850 hPa, from 850 hPa to 500 hPa and finally from 500 to 200 hPa. For this we sum up all grid boxes within the respective area.

- *Abstract, Line 15: Change to: "In comparison with observations, both EMAC and COSMO/MESSy show strengths and weaknesses."*

  Reply: Done.

- *P2L18 – "consistence" should be "consistency"*

  Reply: Done.

- *P4L15 remove second comma:"Thus, it is desirable that all..."*

  Reply: Done.

- *P7L5: Rearrange to: This is very important, especially in mountainous terrain, as COSMO/MESSy resolves the topography much better.*

  Reply: Done.

- *B7L8: suggest adding a comma before the word measurements to make this sentence a bit clearer*

  Reply: Done.

- *Caption fig 5: please specify that the middle row is for June 2008*

  Reply: A note is added

- *P9L28: missing "in"*

  Reply: Done.

- *P12L10: Remove first comma: "In order to check if the vertical distribution of ozone is well simulated,..."*

  Reply: Done.

- *P16L8: Remove first comma: "To investigate if we can..."*

  Reply: Added.

- *P16L16: "In addition, also" – you don't need both of these terms in this sentence*

  Reply: Indeed - wee agree with referee#2.

- *P16L17: "of" should be "off"*

  Reply: Done.

- *P17L15: The word "especially" seems a bit out of place to me here. Suggest "Particularly" instead.*

  Reply: Done.

- *P17L23: Again, "especially" seems out of place here. Suggest "particularly," or "in particular,". Later in this sentence "as by the coarser" should be "than by the coarser"*

  Reply: Done.

- *P17L26: "good" should be "well"*

  Reply: Done.

- *P17L27: Another sentence beginning with especially – perhaps you wish to keep this one, however I'd remove the word as the sentence works as well without it.*

  Reply: Removed.

- *P17L28: Sentence starting with "The comparison" – remove the first comma. The final "to" should be "too".*

  Reply: Done.

- *P17L32: remove first comma*

  Reply: Removed.

**References**

Jöckel, P., Tost, H., Pozzer, A., Kunze, M., Kirner, O., Brenninkmeijer, C. A. M., Brinkop, S., Cai, D. S., Dyroff, C., Eckstein, J., Frank, F., Garny, H., Gottschaldt, K.-D., Graf, P., Grewe, V., Kerkweg, A., Kern, B., Matthes, S., Mertens, M., Meul, S., Neumaier, M., Nützel, M., Oberländer-Hayn, S., Ruhnke, R., Runde, T., Sander, R., Scharffe, D., and Zahn, A.: Earth System Chemistry integrated Modelling (ESCiMo) with the Modular Earth Submodel System (MESSy) version 2.51, Geoscientific Model Development, 9, 1153–1200, 10.5194/gmd-9-1153-2016, http://www.geosci-model-dev.net/9/1153/2016/, 2016.

Taylor, K. E.: Summarizing multiple aspects of model performance in a single diagram, J. Geophys. Res. Atmos., 106, 7183–7192, 10.1029/2000JD900719, http://dx.doi.org/10.1029/2000JD900719, 2001.